| Open Peer Review | Biotechnology | Methods and Protocols
# Targeted rRNA depletion enables efficient mRNA sequencing in diverse bacterial species and complex co-cultures

Kellie A. Heom,[1,2] Chatarin Wangsanuwat,[1,2] Lazarina V. Butkovich,[1] Scott C. Tam,[1] Annette R. Rowe,[3] Michelle A. O'Malley,[1,2] Siddharth S. Dey[1,2,4]

**ABSTRACT**    Bacterial mRNA sequencing is inefficient due to the abundance of ribosomal RNA that is challenging to deplete. While commercial kits target rRNA from common bacterial species, they are frequently inefficient when applied to divergent species, including those from environmental isolates. Similarly, other methods typically employ large probe sets that tile the entire length of rRNAs; however, such approaches are infeasible when applied to many species. Therefore, we present EMBR-seq+, which requires fewer than 10 oligonucleotides per rRNA by combining rRNA blocking primers with RNase H-mediated depletion to achieve rRNA removal efficiencies of up to 99% in diverse bacterial species. Furthermore, in more complex microbial co-cultures between *Fibrobacter succinogenes* strain UWB7 and anaerobic fungi, EMBR-seq+ depleted both bacterial and fungal rRNA, with a fourfold improvement in bacterial rRNA depletion compared with a commercial kit, thereby demonstrating that the method can be applied to non-model microbial mixtures. Notably, for microbes with unknown rRNA sequences, EMBR-seq+ enables rapid iterations in probe design without requiring to start experiments from total RNA. Finally, efficient depletion of rRNA enabled systematic quantification of the reprogramming of the bacterial transcriptome when cultured in the presence of the anaerobic fungi *Anaeromyces robustus* or *Caecomyces churrovis*. We observed that *F. succinogenes* strain UWB7 downregulated several lignocellulose-degrading carbohydrate-active enzymes in the presence of anaerobic gut fungi, suggesting close interactions between two cellulolytic species that specialize in different aspects of biomass breakdown. Thus, EMBR-seq+ enables efficient, cost-effective, and rapid quantification of the transcriptome to gain insights into non-model microbial systems.

**IMPORTANCE**    Microbes present one of the most diverse sources of biochemistry in nature, and mRNA sequencing provides a comprehensive view of this biological activity by quantitatively measuring microbial transcriptomes. However, efficient mRNA capture for sequencing presents significant challenges in prokaryotes as mRNAs are not poly-adenylated and typically make up less than 5% of total RNA compared with rRNAs that exceed 80%. Recently developed methods for sequencing bacterial mRNA typically rely on depleting rRNA by tiling large probe sets against rRNAs; however, such approaches are expensive, time-consuming, and challenging to scale to varied bacterial species and complex microbial communities. Therefore, we developed EMBR-seq+, a method that requires fewer than 10 short oligonucleotides per rRNA to achieve up to 99% rRNA depletion in diverse bacterial species. Finally, EMBR-seq+ resulted in a deeper view of the transcriptome, enabling systematic quantification of how microbial interactions result in altering the transcriptional state of bacteria within co-cultures.

**KEYWORDS**    bacterial mRNA sequencing, rRNA depletion, non-model microbial sequencing, fungal and bacterial co-cultures, lignocellulose deconstruction

Address correspondence to Michelle A. O'Malley, momalley@ucsb.edu, or Siddharth S. Dey, sdey@ucsb.edu.

Kellie A. Heom and Chatarin Wangsanuwat contributed equally to this article. Co-first authors agreed to alternate order of appearance for multiple publications.

The authors declare no conflict of interest.

See the funding table on p. 18.

Microbes remain one of nature's largest and most diverse sources of biochemical transformations. Many bacterial processes, such as their ability to grow on unprocessed feedstocks, are interesting as their mechanisms for releasing sugars or other metabolites could lead to advances in bio-based fuels or other value-added chemicals (1–4). mRNA sequencing (mRNA-seq) provides an opportunity to gain insights into these biochemical reactions as it enables a quantitative and comprehensive view of the transcriptome, which underlies all biological activity. For example, mRNA-seq has been used to quantify the expression of proteins, transporters, and carbohydrate-active enzymes (CAZymes) that are upregulated by *Fibrobacter succinogenes* strain S85 during cellulose utilization compared with when grown on glucose (5). Furthermore, *F. succinogenes* and other cellulolytic strains have been isolated in nature from complex communities, such as the rumen microbiome, and therefore, it is also critical to characterize how interactions between different microbial species mutually tune their expression (6–8). An important aspect of these microbial communities is understanding how different community members utilize different classes of CAZymes to efficiently deconstruct biomass. CAZymes are classified based on biochemical function or sequence homology into six broad families [glycoside hydrolase (GH), glycosyltransferase (GT), polysaccharide lyase (PL), carbohydrate esterase (CE), auxiliary activities (AA), and carbohydrate binding module (CBM)], and the comparative CAZyme expression profiles of community members can help inform the design of engineered communities for bio-based energy solutions (9, 10). Thus, more broadly, efficient sequencing of mRNA from non-model bacteria in microbial consortia will enable a deeper understanding of the role of each member within complex communities and will provide opportunities to advance microbiome engineering for a variety of agricultural, environmental, or energy-related applications.

However, it is more challenging to perform mRNA-seq on bacteria and especially those that exist in co-cultures. This is because mRNA-seq is inherently inefficient in bacteria, due to limitations associated with removing ribosomal RNA. In eukaryotic species, the mRNA contains a poly-A tail at the 3′-end enabling poly-T primers to selectively reverse transcribe the mRNA and not the rRNA (11, 12). However, prokaryotic mRNA does not have a 3′-end poly-A tail, making it more challenging to uniquely capture mRNA compared with rRNA or other non-coding RNA molecules (13, 14). If left undepleted, rRNA can consume up to 90% of the sequencing data with uninformative reads, resulting in highly inefficient sequencing of mRNA (15, 16). Therefore, efficient, high-throughput and cost-effective rRNA depletion is key to obtaining high-resolution maps of gene expression programs in bacterial species.

To accomplish this, several commercial kits for prokaryotic rRNA depletion have been developed, including the Ribo-Zero Plus Microbiome rRNA Depletion Kit (Illumina), RiboMinus Bacteria 2.0 Transcriptome Isolation Kit (Thermo Fisher), and the NEBNext rRNA Depletion Kit for Bacteria (NEB). These kits leverage sequence-specific rRNA removal, which is highly efficient at depleting the target rRNA (Table S1). However, these kits are limited by the breadth of their probe sets, which restricts their application in diverse microbial systems. Furthermore, the cost of these kits typically ranges from approximately \$30 to \$80 per sample. In response to the lack of specificity to varied bacterial species and the high cost, many research groups have also developed methods for rRNA depletion. A majority of these methods employ either biotinylated oligonucleotides that hybridize with rRNA that are then removed using streptavidin-coated paramagnetic beads (17, 18) or oligonucleotides that hybridize with rRNA and act as a guide for RNase H-based enzymatic digestion (19–21). Similarly, CRISPR-Cas9-mediated degradation of rRNA has also been used to sequence bacterial mRNA (22). However, all these methods use large probe sets, typically up to hundreds of probes per bacterial species, to achieve depletion along the full length of each rRNA subunit. An alternative approach to large oligonucleotide or guide RNA arrays is to use the terminator 5′-phosphate-dependent exonuclease, which selectively degrades rRNA containing 5′-monophosphate ends while having minimal activity on mRNA molecules with 5′-triphosphate

ends (23). However, this method has previously been shown to be less efficient at depleting rRNA compared with the other techniques discussed above (24, 25) (Table S1).

To overcome challenges associated with the high cost and the time-intensive activity of developing large probe sets for every new bacterial species profiled, we previously published a method, EMBR-seq (Enriched mRNA by Blocked rRNA) (25), that only requires three to four 30-nt primers per rRNA species (approximately nine primers per bacterial species) for rRNA depletion. EMBR-seq used a combination of 3′-end polyadenylation of prokaryotic mRNA together with specifically designed blocking primers to prevent the downstream amplification of rRNA molecules. In this study, we have developed a significantly improved method, EMBR-seq+, that integrates our previous method with RNase H-based digestion to deplete rRNA at high efficiencies while still employing fewer than 10 oligonucleotides for each rRNA species. In addition, we show that EMBR-seq+ can be easily applied to diverse bacterial species, and compared with EMBR-seq, EMBR-seq+ enables rapid iterations in probe design without having to restart experiments from total RNA to deplete rRNA from microbial species with poorly annotated genomes and unknown rRNA sequences. We demonstrate this in a co-culture system of anaerobic bacteria and fungi native to the rumen microbiome and show that EMBR-seq+ is instrumental in providing deeper coverage of the bacterial transcriptome. In particular, for these bacterial-fungal co-cultures, we found that the RNase H step enabled EMBR-seq+ to outperform both EMBR-seq and RNase H-based depletion alone in terms of rRNA depletion efficiency. Finally, we also investigate transcriptional changes in the bacteria grown under monoculture or co-culture conditions to gain a deeper understanding of how cell-cell interactions between microbial species affect the CAZyme expression profile in these bacteria. In summary, EMBR-seq+ is a highly efficient, high-throughput, and cost-effective platform for sequencing bacterial mRNA that also allows for rapid improvements in probe design to deplete rRNA from poorly annotated and non-model microbial species.

## RESULTS

### EMBR-seq+ combines blocking primers and RNase H digestion to efficiently deplete rRNA

EMBR-seq+ builds off the EMBR-seq method we have previously developed, in which rRNA depletion is achieved using a small set of short blocking primers designed to hybridize at the 3′-end of 16S, 23S, and 5S rRNA in the bacterial species of interest (Fig. 1A). Briefly, in EMBR-seq, total RNA is polyadenylated in the presence of the rRNA blocking primers to preferentially polyadenylate mRNA. The blocking primers not only target the 3′-end of rRNA but also a few "hotspot" locations that are enriched with reads within the rRNA sequence (Table S2). For *E. coli*, the hotspots present in the "No depletion" control library are targeted and efficiently depleted in the other libraries using hotspot-specific blocking primers (see inset in Fig. 1B). Next, a poly-T primer containing a sample barcode, Illumina RA5 sequence, and T7 promoter in its overhang is used to reverse transcribe the polyadenylated molecules. Subsequently, the reverse transcription product is converted to cDNA and then amplified by *in vitro* transcription (IVT) using the T7 polymerase. As a majority of the rRNA molecules are reverse transcribed by the blocking primers instead of the poly-T primers, rRNA-derived cDNA molecules lack a T7 promoter and are therefore excluded from amplification during IVT. The resulting amplified RNA (aRNA) is converted to the final Illumina library by incorporating the Illumina RA3 sequence during a subsequent reverse transcription step followed by addition of the Illumina P5 and P7 adapters during PCR (Fig. 1A).

In EMBR-seq+, we introduced an additional RNase H treatment step to specifically target the remaining rRNA-derived molecules that are present in the aRNA after IVT. In this step, 50-nt ssDNA probes are hybridized near the junction of the Illumina RA5 adapter with the rRNA sequence, with additional probes targeting three to four hotspot locations along the rRNA-derived aRNA molecules. The probes form aRNA-DNA duplexes on the rRNA-derived aRNA molecules which are selectively degraded with RNase H,

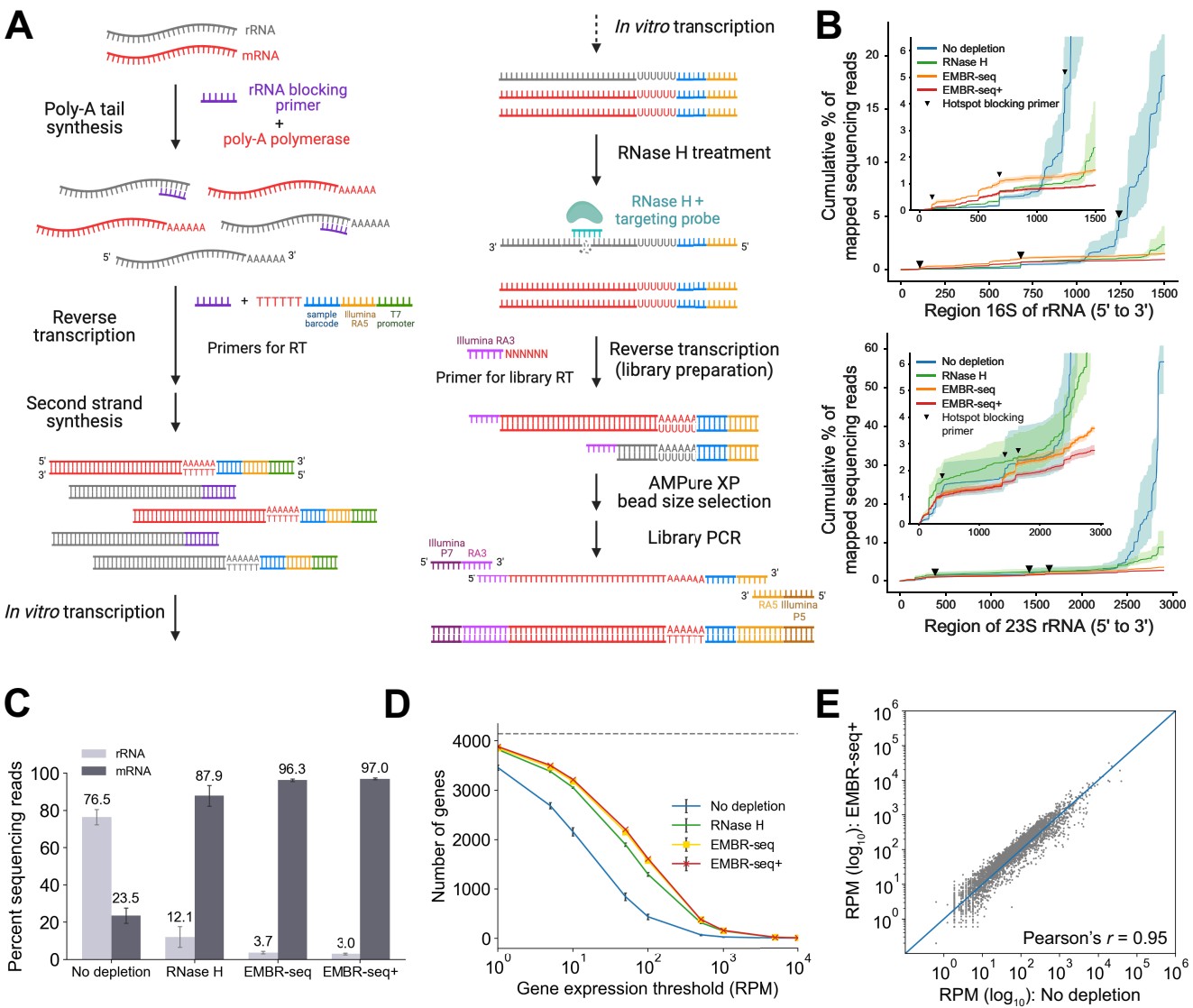

**FIG 1** EMBR-seq+ depletes rRNA from *E. coli* total RNA at high efficiency without introducing technical biases using a small number of rRNA-specific probes. (A) Schematic for EMBR-seq+ shows that a combination of blocking primers and RNase H-based targeting is used to deplete rRNA. This panel was generated using Biorender.com. (B) Cumulative percentage of reads mapping along *E. coli* 16S and 23S rRNA for different depletion methods. Bold lines indicate mean values, and shaded regions indicate the minimum and maximum over three independent experiments. Inverted triangles indicate location of hotspot regions targeted by blocking primers. Insets highlight the depletion of reads from hotspot locations along the 16S and 23S rRNA sequence. (C) Percent of sequencing reads mapping to rRNA and mRNA in *E. coli*. Bars indicate mean values and error bars indicate standard deviations over three independent experiments. rRNA depletion in the "RNase H," "EMBR-seq," and "EMBR-seq+" conditions is all statistically significant from the "No depletion" condition (permutation test, $P < 10^{-5}$). (D) Number of genes detected above different gene expression thresholds for *E. coli*. Points indicate mean values, and error bars indicate standard deviations over three independent experiments. (E) Correlation of gene expression between the "No depletion" and "EMBR-seq+" conditions (Pearson's $r = 0.95$). Reads per million (RPM) is computed after removal of rRNA reads. The *x*- and *y*-coordinates of each point indicate mean values over three independent experiments in the two conditions. For consistent comparison across methods and to control for the variability in sequencing depth across samples, panels B and D show data that have been downsampled to 1 million sequencing reads.

resulting in either the removal of the Illumina RA5 adapter at the 5′ end of these molecules or the generation of very short fragments that are excluded during the downstream PCR amplification or size selection steps, respectively, thereby preventing their incorporation into the final Illumina library. As the RNase H treatment is performed on aRNA molecules to cleave the partial RA5 Illumina adapter, EMBR-seq+ requires only three to four ssDNA probes per rRNA species (approximately nine per bacterial species)

compared with most other RNase H-based depletion methods that typically need to tile the entire rRNA molecules with up to hundreds of ssDNA probes per bacterial species (17–22). After RNase H treatment, the rRNA-depleted aRNA is reverse transcribed and then amplified with PCR to obtain the final Illumina libraries (Fig. 1A). We tested two enzymes and reaction temperatures: (i) Invitrogen RNase H at 16°C and (ii) Hybridase Thermostable RNase H at 45°C. We found that for both the "RNase H" only and EMBR-seq+ libraries, the rRNA depletion efficiency was very similar with the two RNase H enzymes and decided to proceed with the Invitrogen RNase H for all subsequent experiments (Fig. S1).

As proof-of-principle, we first applied these methods to *E. coli* where rRNA was depleted to 12.1% of the sequenced reads when treated with RNase H alone and to 3.7% with EMBR-seq and 3% with EMBR-seq+, suggesting that the combined use of blocking primers and RNase H depletion in EMBR-seq+ can remove most of the rRNA-derived molecules from the sequencing libraries (Fig. 1C). This efficiency of rRNA depletion in EMBR-seq+ is similar to that achieved by other methods and commercial kits (Table S1). In the libraries in which only RNase H was used for rRNA depletion, no blocking primers were added during the poly-A tail synthesis and initial reverse transcription step. Similarly, in the "No depletion" control, neither blocking primers nor RNase H depletion was performed. Furthermore, as expected, for the same sequencing depth, increased depletion of rRNA in EMBR-seq+ resulted in the detection of the most unique genes above every expression threshold over three orders of magnitude (Fig. 1D). Finally, gene expression levels quantified by EMBR-seq+ were highly correlated to the "No depletion" libraries (Pearson $r = 0.95$), suggesting that the use of blocking primers and RNase H in EMBR-seq+ does not introduce technical biases (Fig. 1E; Table S3). Similarly, the transcriptomes obtained from EMBR-seq and EMBR-seq+ were highly correlated, suggesting that RNase H-based depletion in EMBR-seq+ does not introduce technical biases (Table S3). Collectively, these experiments show that EMBR-seq+ can efficiently deplete rRNA from bacterial total RNA without introducing technical artifacts to quantify the transcriptome at high depth.

## EMBR-seq+ can be easily applied in distant prokaryotic species

EMBR-seq+ requires fewer than 20 ssDNA oligonucleotides per bacterial species (fewer than 10 blocking primers and fewer than 10 probes for RNase H depletion) to deplete 16S, 23S, and 5S rRNA, and therefore, it is easy and inexpensive to customize the method for any species of interest (Table S2). We tested EMBR-seq+ on *Fibrobacter succinogenes* strain UWB7, a strain of cellulolytic bacteria isolated from cow rumen, and *Geobacter metallireducens*, a strain that reduces Fe(III) oxide and can exhibit syntropy with methanogens via direct interspecies electron transfer (26–28). When employing EMBR-seq, rRNA was depleted to 10.3% in *F. succinogenes* strain UWB7 and 3.1% in *G. metallireducens* while EMBR-seq+ depleted rRNA to 7.6% in *F. succinogenes* strain UWB7 and 1.1% in *G. metallireducens*, highlighting the improved efficiency of rRNA depletion when combining blocking primers and RNase H digestion in EMBR-seq+ (Fig. 2A). Furthermore, as with *E. coli*, custom-designed blocking primers and RNase H probes were key to efficiently depleting rRNA reads from hotspot loci observed in the "No depletion" controls for both *F. succinogenes* strain UWB7 and *G. metallireducens* (Fig. 2B; Fig. S2A). Consistent with the improved rRNA depletion, an increased number of unique genes was detected in EMBR-seq+ above every expression threshold for both *F. succinogenes* strain UWB7 and *G. metallireducens* (Fig. 2C; Fig. S2B). Finally, when compared with the "No depletion" controls, the blocking primers and RNase H probes did not introduce technical bias in the quantification of gene expression in EMBR-seq+ (Pearson $r = 0.94$ and $r = 0.97$ for *F. succinogenes* strain UWB7 and *G. metallireducens*, respectively) (Fig. 2D; Fig. S2C; Table S3). These results show that the design of a small number of short probes in EMBR-seq+ can be used to easily extend the method with high efficiency to varied bacterial species.

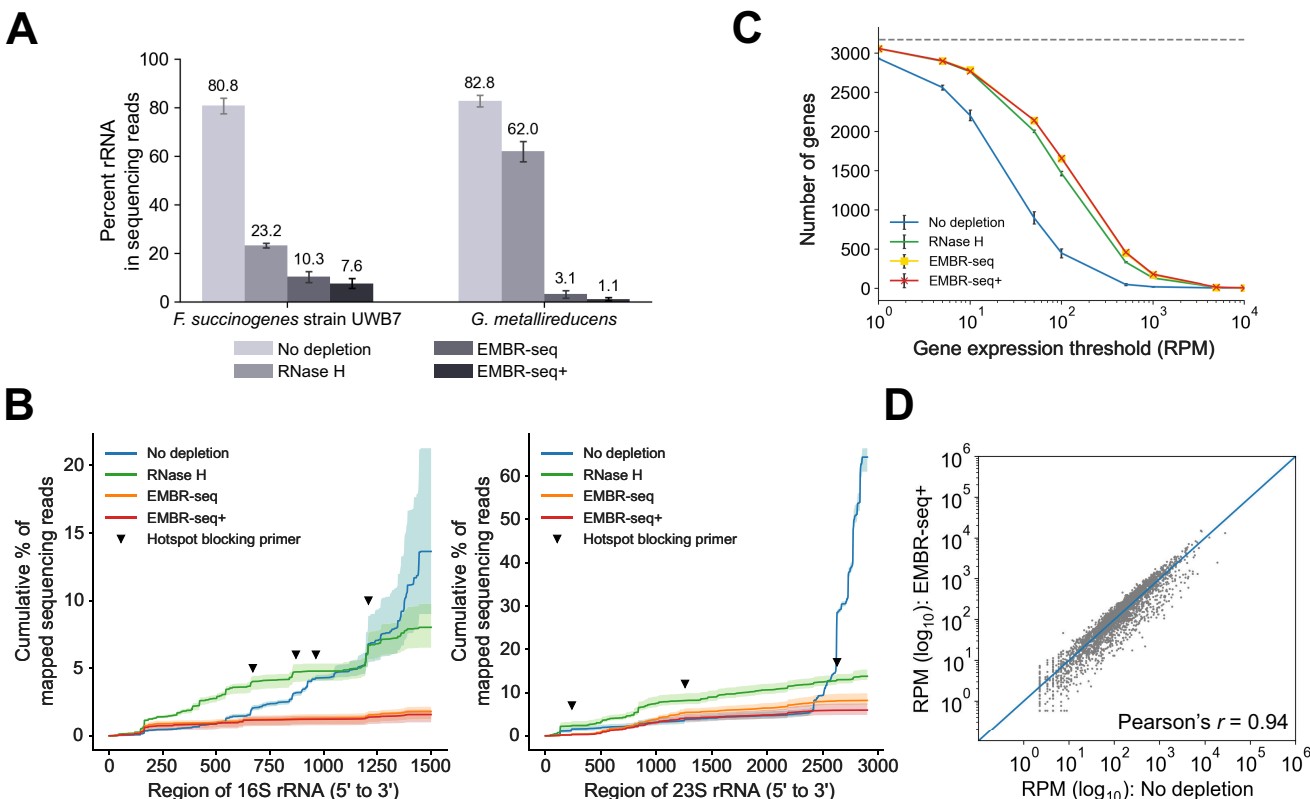

**FIG 2** EMBR-seq+ can be easily extended to diverse bacterial species. (A) Percent of sequencing reads mapping to rRNA in *F. succinogenes* strain UWB7 and *G. metallireducens* for different rRNA depletion methods. Bars indicate mean values and error bars indicate standard deviations over three independent experiments. rRNA depletion in the "RNase H," "EMBR-seq," and "EMBR-seq+" conditions is statistically significant from the "No depletion" condition (permutation test, $P < 10^{-5}$). (B) Cumulative percentage of reads mapping to 16S and 23S rRNA of *F. succinogenes* strain UWB7 for different rRNA depletion methods. Bold lines indicate mean values, and shaded regions indicate the minimum and maximum over three independent experiments. Inverted triangles indicate the location of hotspots targeted by blocking primers. (C) Number of genes detected above different gene expression thresholds for *F. succinogenes* strain UWB7. Points indicate mean values, and error bars indicate standard deviations over three independent experiments. (D) Correlation of gene expression between the "No depletion" and "EMBR-seq+" conditions for *F. succinogenes* strain UWB7 (Pearson's $r = 0.94$). RPM is computed after removal of rRNA reads. The $x$- and $y$-coordinates of each point indicate mean values over three independent experiments in the two conditions. For consistent comparison across methods and to control for the variability in sequencing depth across samples, panels B and C show data that have been downsampled to 1 million sequencing reads.

## EMBR-seq+ enables efficient prokaryotic mRNA sequencing in a fungal-bacterial co-culture system

Next, we aimed to apply EMBR-seq+ to more complex systems that mimic microbial communities in nature, such as co-cultures of bacteria with different fungal species. As an example, anaerobic organisms isolated from the rumen microbiome are particularly interesting because of their ability to grow on raw, unprocessed substrates such as grass and depolymerize lignocellulose, a feature attributed to carbohydrate-active enzymes found in their genomes. *F. succinogenes* is an extensively studied cellulolytic bacterial species, because it is frequently identified in rumen samples and found to encode a large number of CAZymes in its genome (5, 6, 26, 29–32). More recently, anaerobic gut fungal genomes have also been discovered to be exceptionally abundant with CAZymes (33–35). As these species exist in consortia in nature rather than in isolation, understanding interactions between different species is key to gaining insights into their function in native environments and for engineering stable microbial systems *in vitro* (36–38). A previous study examined transcriptomic changes to the anaerobic gut fungi *Anaeromyces robustus* or *Caecomyces churrovis* when cultured together with *F. succinogenes* strain UWB7 compared with monoculture conditions and identified several novel

components of the fungal secondary metabolism that were upregulated in co-culture (37). This study focused primarily on the remodeling of the fungal transcriptome induced by the presence of bacteria; however, this work did not characterize transcriptional changes in the bacteria *F. succinogenes* strain UWB7 in co-culture with *A. robustus* due to two technological limitations associated with (i) the low abundance of bacterial RNA compared with the fungi and 2(ii) the lack of efficient bacterial rRNA depletion.

To overcome these limitations, we focused on capturing the bacterial transcriptome by applying EMBR-seq+ to two independent co-cultures of *F. succinogenes* strain UWB7 with different anaerobic gut fungal partners, *A. robustus* or *C. churrovis* (34, 39, 40). In our initial co-culture experiments with *F. succinogenes* strain UWB7 and *A. robustus*, we applied EMBR-seq to total RNA extracted from co-culture pellets [denoted as "Unprocessed total RNA (EMBR-seq)"] to find only 11.8% of the mapped reads derived from the bacteria with the remaining 88.2% of reads derived from the fungi (Fig. 3A and B). We hypothesized that this large excess of fungal reads in our data set is derived from an initial overabundance of fungal RNA in the co-culture total RNA pellets and a lack of targeting to deplete fungal RNA in EMBR-seq+. As our goal was to characterize the transcriptome of *F. succinogenes* strain UWB7 within the co-culture at high resolution, we aimed to deplete the fungal mRNA and rRNA. We first used poly-T magnetic beads to deplete polyadenylated fungal mRNA prior to applying EMBR-seq, resulting in a library [denoted as "Poly-A depleted (EMBR-seq)"] with 24.9% bacteria reads.

To increase the fraction of bacterial reads beyond the 2.1-fold achieved over "Unprocessed total RNA (EMBR-seq)," we next attempted to deplete the fungal rRNA. However, as with many non-model microbes, a major challenge was that the rRNA sequences were not annotated for either of the fungal genomes. Therefore, we first mapped the sequencing data from the "Poly-A depleted (EMBR-seq)" sample against the reference genome scaffolds and identified a series of adjacent pileups of mapped reads on the same scaffold that were spaced apart according to the expected sizes of the 18S, 28S, and 5.8S eukaryotic rRNA subunits (Fig. 3C). Thus, we inferred that the three genomic loci on this scaffold correspond to the fungal rRNA subunits and designed five RNase H probes to specifically deplete the RNA transcribed from these locations (Table S2). We also aligned our probe sequences against the rRNA annotations of another closely related fungal species, *Orpinomyces* sp. OUS1, and confirmed that they align to the 3′-ends of 18S, 28S, and 5.8S rRNA (41). By combining the upstream poly-A bead-based fungal mRNA depletion prior to EMBR-seq+ with the RNase H-based fungal rRNA depletion during EMBR-seq+ [denoted as "Poly-A depleted & RNase H treated (EMBR-seq+)"], we were able to successfully increase the percentage of bacterial reads in the final sequencing libraries to 56.8%, a 1.75-fold improvement over previous work (Fig. S3A) (37). We hypothesize that we still obtained reads mapping to the fungal genome due to the following reasons. First, the RNase H probes to deplete fungal rRNA were designed based on putative rRNA sequences resulting from our scaffold mapping-based approach. The lack of experimentally validated rRNA annotation may have led to suboptimal RNase H probe designs and less efficient rRNA depletion. Second, our poly-T pull down may not have captured a small fraction of fungal mRNA, resulting in these molecules being sequenced. Third, we mapped our sequencing reads to the entire fungal genome, rather than the coding transcriptome as these genomes are not well annotated resulting in transcriptome references that are likely incomplete. However, a consequence of this, together with the poly-adenylation step in EMBR-seq+, is that we potentially also sequenced and mapped non-polyadenylated non-coding RNA. Nevertheless, our combined fungal RNA depletion approach enabled us to effectively enrich bacterial RNA from our co-cultures. Furthermore, we also applied these two fungal depletion treatments to the co-culture of *F. succinogenes* strain UWB7 with *C. churrovis* to observe a similar enrichment of bacteria-derived reads to 41.3% in EMBR-seq+, suggesting that this is a robust approach to quantify the bacterial transcriptome in complex microbial communities (Fig. S4 and S5). More generally, the fungal rRNA depletion strategy demonstrates that the RNase H digestion component of EMBR-seq+

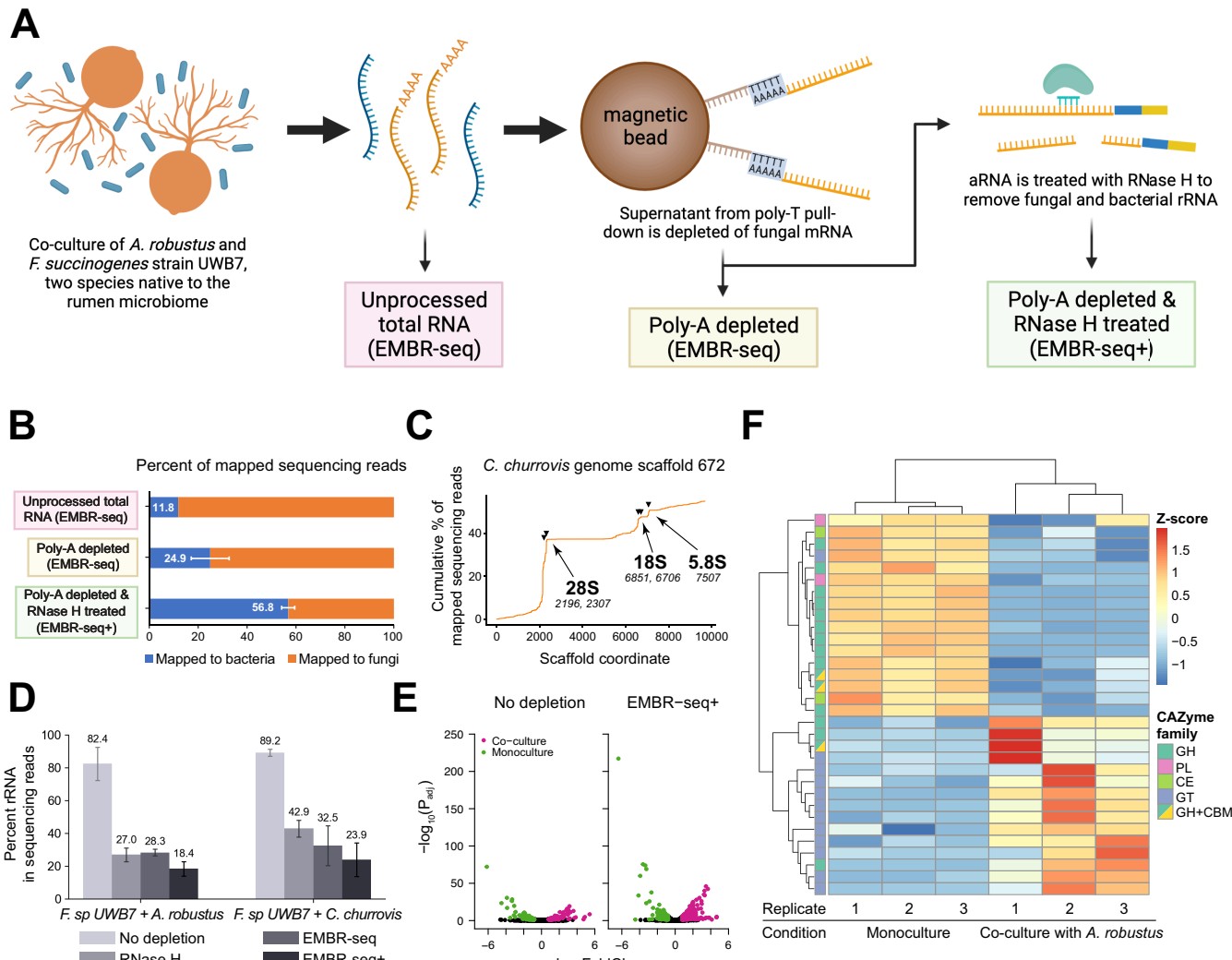

**FIG 3** EMBR-seq+ efficiently depletes rRNA of *F. succinogenes* strain UWB7 grown in co-culture with *A. robustus*. (A) Schematic of fungal mRNA and rRNA removal from co-culture-derived total RNA samples for more efficient bacterial mRNA sequencing. This panel was generated using Biorender.com. (B) Percentage of reads mapping to the bacterial and fungal genomes using the three strategies described in panel A: total RNA isolated from co-culture pellets treated with EMBR-seq to remove bacterial rRNA ["Unprocessed total RNA (EMBR-seq)"], fungal poly-adenylated mRNA-depleted total RNA treated with EMBR-seq to remove bacterial rRNA ["Poly-A depleted (EMBR-seq)"], and "Poly-A depleted (EMBR-seq)" library additionally treated with RNase H to remove fungal and bacterial rRNA ["Poly-A depleted & RNase H treated (EMBR-seq+)"]. For the "Unprocessed total RNA (EMBR-seq)" library, $n = 1$. For "Poly-A depleted (EMBR-seq)" and "Poly-A depleted & RNase H treated (EMBR-seq+)" libraries, $n = 3$. Bars indicate mean values, and error bars indicate standard deviation from the mean. (C) Cumulative percentage of sequencing reads from the "Poly-A depleted (EMBR-seq)" library that align to the *C. churrovis* genome scaffold 672. The three rRNA hotspots were inferred to be the 3′-ends of 18S, 28S, and 5.8S rRNA for *C. churrovis*. Black inverted triangles and italicized text indicate target locations of the five RNase H probes used for fungal rRNA depletion in the EMBR-seq+ libraries. (D) Percent of sequencing reads mapping to the rRNA of *F. succinogenes* strain UWB7 for different depletion methods when grown in co-culture with *A. robustus* or *C. churrovis*. All four samples are poly-A depleted to remove fungal mRNA. Bars indicate mean values, and error bars indicate standard deviation over three independent experiments. For co-cultures of *F. succinogenes* strain UWB7 and *A. robustus*, rRNA depletion in the "RNase H," "EMBR-seq," and "EMBR-seq+" conditions is statistically significant from the "No depletion" condition (permutation test, $P < 0.05$). For co-cultures of *F. succinogenes* strain UWB7 and *C. churrovis*, rRNA depletion in the "RNase H," and "EMBR-seq+" conditions is statistically significant from the "No depletion" condition (permutation test, $P < 0.05$). (E) Volcano plot of differentially expressed genes of *F. succinogenes* strain UWB7 grown in monoculture vs. co-culture with *A. robustus*, for the "No depletion" condition and after rRNA depletion using EMBR-seq+. $n = 3$ for both culture conditions and for both the library preparation conditions. Colored points indicate differentially expressed genes, determined by thresholds of $|\log_2 (\text{fold change})| > 0.8$, $P_{adj} < 0.05$, and RPM > 2. (F) Heatmap of differentially expressed CAZymes in *F. succinogenes* strain UWB7 grown in monoculture vs. co-culture with *A. robustus*. Genes are colored on the left side of the heatmap based on the CAZyme family they are associated with.

enables rapid iterations of probe designs; this step is implemented on a small amount of amplified aRNA, an intermediate product in EMBR-seq+, thereby offering a convenient approach to circumvent the need to prepare new libraries from total RNA or re-grow the co-cultures.

After successfully enriching for bacterial RNA from bacteria-fungal co-cultures, we next quantified the depletion of bacterial rRNA within these co-cultures. As before, we observed that EMBR-seq+ resulted in the highest depletion of *F. succinogenes* strain UWB7 rRNA compared with "RNase H" or "EMBR-seq" for both co-culture systems, demonstrating that the combination of blocking primers and RNase H digestion is highly effective in depleting bacterial rRNA (Fig. 3D). Furthermore, the rRNA depletion efficiency of EMBR-seq+ was significantly improved over a previous transcriptomic study using the same co-culture pairings (5.1- and 3.9-fold increased rRNA depletion for co-cultures with *A. robustus* and *C. churrovis*, respectively) (Fig. S3). Similarly, compared with the other methods, we observed that EMBR-seq+ detected the most genes above all expression thresholds and did not introduce technical bias in quantifying gene expression levels (Fig. S6). Overall, these results show that EMBR-seq+ can be easily adapted to efficiently enrich for bacterial mRNA from complex microbial communities.

## Differential gene expression between monocultures and co-cultures

After establishing that EMBR-seq+ can provide a detailed view of bacterial gene expression programs within co-cultures, we next wanted to characterize the transcriptional response of *F. succinogenes* strain UWB7 in the presence of the two different strains of anaerobic fungi. To accomplish this, we used DESeq2 to perform differential gene expression analysis for *F. succinogenes* strain UWB7 grown in monoculture vs. co-culture. While 140 genes were upregulated for *F. succinogenes* strain UWB7 in monoculture and 128 genes were upregulated in co-culture with *A. robustus* for the "No depletion" data sets, we identified 224 upregulated genes in monoculture and 374 upregulated genes in co-culture for the "EMBR-seq+" data sets (Fig. 3E, Fig. S7 and S8). We observed similar robust results for experiments with *C. churrovis*, with 113 genes upregulated in monoculture and 158 genes upregulated in co-culture for the "No depletion" data sets, while bacterial RNA enriched and rRNA depleted data sets obtained from EMBR-seq+ displayed 247 upregulated genes in monoculture and 149 upregulated genes in co-culture (Fig. S7 and S9a; Table S4). While the relative trends between monoculture and co-culture conditions were consistent, we observed that the biological replicates for the co-cultures showed some variation, possibly due to the increased complexity of the interactions between species that might depend on their relative abundances. Therefore, to ensure consistency, all the monoculture and co-culture replicates were prepared from the same seed cultures of *A. robustus*, *C. churrovis*, and *F. succinogenes* strain UWB7, and we followed the same experimental timeline for all the experiments.

To better understand the deconstruction of carbon substrates by *F. succinogenes* strain UWB7, we used dbCAN2 (42, 43) to predict CAZyme annotations in the bacterial transcriptome. Within dbCAN2, we performed three predictive searches: (i) HMMER against the dbCAN database, (ii) DIAMOND against the CAZy database, and (iii) HMMR against the dbCAN_sub database. One hundred ninety-one genes were predicted to be CAZymes in at least two out of the three tools, and we found that 189 of these genes were expressed with an RPM of at least 2 in all the monocultures and co-culture replicates. Notably, in co-cultures with both *A. robustus* and *C. churrovis*, CAZymes represented a substantial fraction of the differentially regulated genes. For example, of the 191 CAZyme genes, 32 (17%) were differentially expressed, with 17 upregulated in monoculture and 15 upregulated in co-culture with *A. robustus* (Fig. 3F). In addition to CAZymes, *F. succinogenes* strain UWB7 also upregulated several molecular chaperones and components of toxin-antitoxin systems when grown in co-culture with either fungal species (Table S5). Finally, many of the transporters found by Swift et al. to be upregulated in co-culture were also upregulated in our EMBR-seq+ data sets (Table S6) (37).

Overall, this expression profile highlights elements of stress response in the bacteria in the presence of the anaerobic gut fungi.

Interestingly, in monoculture and co-culture, the bacteria upregulated different subclasses of CAZymes. In monoculture, *F. succinogenes* strain UWB7 upregulated a number of lignin-active CAZymes in the glycoside hydrolase, polysaccharide lyase, carbohydrate esterase, and carbohydrate binding module subclasses (Fig. 3F; Table S7). Conversely, in co-cultures, the bacteria did not upregulate as many CAZymes in these classes and instead upregulated CAZymes in the glycosyl transferase subclass. While the GH, PL, and CE classes represent enzymes that hydrolyze or cleave different types of bonds found in lignocellulose, the functions of GT enzymes include forming new glycosidic bonds to modify or polymerize sugars (9). The GT enzymes found to be upregulated in the bacteria in co-culture have predicted functions for cell wall biosynthesis, suggesting that the cells could be incorporating sugars in their peptidoglycan layers. Thus, the dramatic transcriptomic shift away from lignocellulose breakdown observed in *F. succinogenes* strain UWB7 in the presence of anaerobic fungi highlights that the interaction between these two species, which are both known to be efficient lignocellulose degraders, substantially alters the bacterial response. Furthermore, out of the thousands of microbial members identified in the cattle rumen, members of the Fibrobacteres phylum have been found to devote the largest proportion of their proteome towards carbohydrate metabolism; therefore, the downregulation of several lignocellulose-modifying CAZymes in the presence of fungi potentially suggests that *A. robustus* releases sugars in such great excess that the *F. succinogenes* strain UWB7 redirects their efforts away from sugar hydrolysis (6, 44). Overall, these findings suggest that *F*. succinogenes strain UWB7 may not need to reach their full lignocellulose-degrading potential in this co-culture system, and from an engineering perspective, a more efficient bioreactor might pair either the *F. succinogenes* strain UWB7 or the *A. robustus* with another organism that has complementary behavior, such as the ability to ferment sugars or produce fatty acid products (10, 45, 46).

In the co-culture experiments with *C. churrovis*, a similar but less pronounced shift in CAZyme expression profile was observed for *F. succinogenes* strain UWB7 (Table S7; Fig. S9B). For example, compared with co-culture experiments with *A. robustus*, fewer GH CAZymes were upregulated in monoculture and a few GH genes were also upregulated in co-culture. These observations potentially reflect that co-cultures with different fungal strains result in altered substrate availability for *F. succinogenes* strain UWB7. This is consistent with recent work showing that *C. churrovis* does not release glucose and xylose in concentrations nearly as high as *A. robustus*, and the lower levels of excess reducing sugars could result in distinct signaling and transcriptome remodeling in the bacteria (46). This is likely due to the fact that *C. churrovis* lacks the rhizoid system found in most other anaerobic fungi, which limits its ability to degrade fibrous biomass (39, 40). Additionally, *F. succinogenes* strain UWB7 in co-culture upregulated a few xylan esterase genes CE6 and CBM+CE6 fusion genes. As these genes were not differentially expressed in co-culture with *A. robustus*, this result agrees with previous findings that *C. churrovis* is unique from other anaerobic fungi because it is a xylan-degrading specialist (39, 47, 48). Fibrobacteres do not have genes for xylan transport or metabolism, and previous work has suggested that their xylan machinery is used to gain access to cellulose (26); therefore, our result suggests a relationship wherein when grown on switchgrass with *C. churrovis*, there is either a direct or indirect benefit to the bacteria for expressing xylan esterase genes. These transcriptional responses in the bacteria highlight how high-quality mRNA-seq from EMBR-seq+ provides a detailed perspective of complex co-culture systems and contributes to our understanding of how these species interact both with each other and also with the substrate present.

## DISCUSSION

In this study, we have developed EMBR-seq+, a method that depletes rRNA in prokaryotic species at high efficiencies to enable cost-effective and improved mRNA sequencing.

To accomplish this, EMBR-seq+ uses a combination of selective polyadenylation on mRNA molecules, blocking primers that prevent downstream amplification of rRNA molecules, and RNase H digestion targeted at rRNA-derived amplified RNA. Importantly, the method is designed to only require a single blocking primer per rRNA sequence at the 3′-end, together with three to four additional blocking primers at "hotspots" and three to four probes to direct RNase H digestion of rRNA. By requiring blocking primers and RNase H probe targets at only a few strategic locations, EMBR-seq+ is significantly easier to implement and less expensive and can be rapidly adapted to any bacterial species, compared with alternate methods that require large primer sets to tile the entire length of rRNA molecules (19–22).

A major change that we introduced in EMBR-seq+ compared with EMBR-seq is the inclusion of RNase H digestion downstream of IVT amplification. A key advantage of the RNase H treatment in EMBR-seq+ over other RNase H-based methods is that it is performed on IVT-amplified RNA rather than the starting total RNA; therefore, if the sequencing results from a previous run indicate the continued presence of one or more highly abundant rRNA fragments, additional RNase H probes can be designed and applied to the same IVT-amplified RNA without having to start again from total RNA, thereby significantly reducing experimental time. We demonstrate this approach in the depletion of fungal rRNA in the co-culture experiments (Fig. 3). Additionally, while many early bacterial single-cell RNA-seq methods lacked an rRNA depletion step, more recent reports employ CRISPR-Cas9-mediated rRNA depletion or the use of probes to only capture mRNA transcripts of interest (49–53). However, as the RNase H-based digestion strategy described in this work requires only a few probes per rRNA species, in the future, it could readily be integrated with single-cell RNA-seq methods that involve IVT-based amplification (54–57). In addition, the use of blocking primers to deplete rRNA can potentially also be scaled down to single cells as we previously showed that EMBR-seq efficiently depletes rRNA from as little as 0.02 ng input total RNA (25). This corresponds to a pool of ~200 single bacteria (assuming ~0.1 pg of RNA per bacterial cell) and is in line with the sensitivity requirements of the existing methods that employ combinatorial barcoding strategies (49, 51, 56, 57). Therefore, we anticipate that in the future EMBR-seq+ could potentially be applied to single cells, thereby adding to the growing toolkit for profiling individual bacterial cells.

In this study, we also showed that EMBR-seq+ can be applied to complex microbial communities, such as a co-culture system of anaerobic bacteria and fungi native to the rumen microbiome to deplete rRNA efficiently from both species. In a previous study, the Ribo-Zero Gold rRNA Removal Kit (Epidemiology) was used for rRNA depletion from the same bacterial and fungal co-cultures; however, rRNA reads contributed to more than 90% of the sequencing data, thereby requiring significantly higher sequencing depths to obtain sufficient coverage of the fungal and bacterial transcriptomes (37). This reduced depletion efficiency potentially arises from the kit's lack of specificity towards non-model microbes due to sequence divergence, highlighting the importance of methods such as EMBR-seq+ that can be rapidly adapted to any species. Furthermore, in the aforementioned study, co-cultures of *A. robustus* and *F. succinogenes* strain UWB7 were sequenced to a depth of ~15 million reads to obtain only 20,000 bacterial mRNA reads, making it impractical to perform differential expression analysis on the bacterial RNA. In contrast, EMBR-seq+ libraries were sequenced to a depth of ~12 million reads to obtain ~1.9 million *F. succinogenes* strain UWB7 mRNA reads, representing a dramatic improvement in quantifying the bacterial transcriptome (Fig. S3). Overall, this demonstrates that the small number of probes required in EMBR-seq+ enables rapid application of this method to varied microbial species while retaining high rRNA depletion efficiency.

Finally, to gain new insights into how *F. succinogenes* strain UWB7 is affected in co-culture with anaerobic gut fungi, we used EMBR-seq+ to quantify transcriptional changes in the bacteria when transitioned from monoculture to co-culture conditions. In previous work, Swift et al. designed this co-culture system for the bacteria to exert pressure on the fungi and induce the production of fungal secondary metabolites;

however, we found that the bacteria also exhibit a substantial stress response and changes to its gene expression program (37). We observed a transcriptional shift in the bacterial CAZyme profile, marked by a decrease in the expression of lignocellulose-active CAZymes in the presence of the anaerobic fungi. In agreement with this, a previous study showed that microbiome samples enriched in anaerobic fungi (including *A. robustus*) release excess reducing sugars in the growth supernatant that potentially acts as a nutrient source for other organisms in its environment (44, 46). Similarly, this is consistent with another previous finding where in fungal-dominated microbiome cultures (achieved by antibiotic treatment), antibiotic-resistant bacteria were found to survive on reed canary grass, despite the absence of CAZymes in their genomes (47). Thus, in general, efficient rRNA depletion using EMBR-seq+ enables a deeper view of the bacterial transcriptome to gain functional insights into the response of bacteria in complex cultures. Given the low cost, ease of design and efficiency of our method, we anticipate that high-throughput screens employing EMBR-seq+ will offer a strategy for rapidly understanding microbial interactions in diverse environmental systems.

## MATERIALS AND METHODS

### *E. coli* cell culture

Overnight cultures of *Escherichia coli MG1655* (ATCC: 700926) were inoculated into fresh LB medium at 1:1,000 and grown at 37°C with shaking (225 rpm). After 4 h, the cells reached the exponential growth phase (verified by $OD_{600}$ measurement) and the culture was centrifuged at $3,000 \times g$ and 4°C for 10 min. The media were removed, and the pellet was resuspended in PBS to a concentration of $10^7$ cells/μL. For RNA extraction, $5 \times 10^7$ cells were added to 750 μL TRIzol, mixed, and stored at −20°C.

### *G. metallireducens* culture

Cultures of *Geobacter metallireducens* GS-15 (DSM 7210) were grown at 30°C in an anaerobic basal salt media (1.3 mM KCl, 4.6 mM $NH_4Cl$, 4.4 mM $NaH_2PO_4$, and 30 mM $NaHCO_3$) amended with 56 mM iron citrate and 20 mM acetate. Wolf's vitamin mix (from DSMZ protocol 141) and metal reducer mineral mix (described by Bretschger et al.) were added from 100× stock to 1× final concentrations (58). All cultures were grown in 20 mL volumes in anaerobic serum vials from 1% culture transfers. Cultures were maintained in DMSO at −80°C when not in use. For harvesting the cells, 1 mL of culture was centrifuged at $8,000 \times g$ and 4°C for 5 min to obtain a pellet. The media was removed and the pellet was resuspended in 750 μL of TRIzol, mixed, and stored at −20°C.

### *F. succinogenes* strain UWB7 monoculture and co-culture with anaerobic gut fungi

*Fibrobacter succinogenes* strain UWB7 monoculture and co-culture with anaerobic gut fungi were prepared based on the experimental protocol described by Swift et al. (37). *F. succinogenes* strain UWB7 was previously gifted by Garret Suen at the University of Wisconsin-Madison, with isolation of the strain described by Neumann and Suen (59). The anaerobic gut fungi *Anaeromyces robustus* and *Caecomyces churrovis* were previously isolated via reed canary grass enrichment from fecal pellets of a Churro sheep at the Santa Barbara Zoo (33, 34, 39).

 *F. succinogenes* strain UWB7, *A. robustus*, and *C. churrovis* were cultivated at 39°C using standard anaerobic techniques (60). Cultures were prepared with a $CO_2$ headspace in Hungate tubes containing 9.0 mL of a modified formulation (MC-) of complex medium C (61), containing 0.25 g/L yeast extract, 0.5 g/L Bacto Casitone, and 7.5 vol% clarified rumen fluid, along with 0.1 g/L of substrate: Avicel PH-101 (Sigma-Aldrich, St. Louis, MO) or milled switchgrass (gift from the U.S. Department of Agriculture). Avicel PH-101 substrate was used for bacterial passaging, whereas switchgrass substrate was used for fungal passaging. After autoclaving, the MC- medium was supplemented with 0.1% (vol/

vol) vitamin solution (ATCC, catalogue no. MD-VS) per 10 mL of final culture volume, as described by Teunissen (62). Each passaging Hungate tube was inoculated with 1.0 mL of cryostock or live inoculum culture, with cultures passaged anaerobically using the sterile syringe needle technique every 2–4 days.

In order to inoculate the *F. succinogenes* strain UWB7 monocultures and co-cultures, seed cultures of *F. succinogenes* strain UWB7, *A. robustus*, and *C. churrovis* were prepared. The *F. succinogenes* strain UWB7 seed culture was prepared by adding 1.0 mL of passaging tube inoculum to a serum bottle containing 40 mL of MC- and 0.4 g of Avicel PH-101. The seed culture of *F. succinogenes* strain UWB7 was incubated at 39°C for 24 h for co-culture inoculum and incubated for a total of 48 h for UWB7 monoculture inoculum. *A. robustus* and *C. churrovis* seed cultures were prepared by adding 1.0 mL of appropriate passaging tube inoculum to serum bottles containing 40 mL of MC- and 0.4 g of switchgrass. The fungal seed cultures were incubated at 39°C for 24 h before 1.0 mL inocula was used to inoculate co-culture Hungate tubes with 8.0 mL MC- and 0.1 g switchgrass ($t = 0$ h) in triplicate. The gut fungi were allowed to grow at 39°C for 24 h ($t = 24$ h) before 1.0 mL of *F. succinogenes* strain UWB7 seed inoculum was introduced. The *A. robustus-F. succinogenes* strain UWB7 co-cultures and *C. churrovis-F. succinogenes* strain UWB7 co-cultures were incubated for an additional 48 h ($t = 72$ h) before samples were harvested for RNA-seq. These experiments were performed in triplicate. *F. succinogenes* strain UWB7 monocultures were prepared by adding 1.0 mL seed culture inocula to Hungate tubes with 9.0 mL MC- and 0.1 g switchgrass. *F. succinogenes* strain UWB7 monocultures were then incubated at 39°C for 24 h before samples were harvested for RNA-seq. This experiment was performed in triplicate. The final contents of each co-culture and monoculture Hungate tube were transferred to 15 mL Falcon tubes (Fisher Scientific, Waltham, MA) and centrifuged at 3,200 × $g$ and 4°C using a swinging-bucket rotor (Eppendorf A-4-81) for 5 min. Supernatants were decanted, and 1.0 mL of RNA preservative was added (63, 64). Cell pellets were flash frozen and stored at −80°C until RNA extraction.

## TRIzol RNA extraction for *E. coli* and *G. metallireducens* cultures

TRIzol (Thermo Fisher Scientific, Cat. # 15596018) RNA extractions were performed following the manufacturer's protocol. Briefly, 750 µL TRIzol containing cell lysates were thawed and mixed with 150 µL chloroform. After centrifugation, the clear aqueous layer was recovered and precipitated with 375 µL of isopropanol and 0.67 µL of GlycoBlue (Thermo Fisher Scientific, Cat. # AM9515). The pellet was washed twice with 75% ethanol, and after the final centrifugation, the resulting pellet was air dried and resuspended in nuclease-free water.

## Liquid N$_2$ grinding and RNeasy RNA extraction for *F. succinogenes* strain UWB7 monoculture and co-culture

For RNA extraction, cell pellets were thawed on ice and centrifuged at 3,200 × $g$ and 4°C using a swinging-bucket rotor for 10 min. RNA was extracted from *F. succinogenes* strain UWB7 monoculture and co-culture pellets using the RNeasy Mini Kit (Qiagen, Germantown, MD) following the protocol for purification of total RNA from plant cells and tissues and filamentous fungi, using the liquid N$_2$ method of cell lysis, QIAshredder homogenization, on-column DNase I digestion, and elution in 30 µL of RNase-free water.

## EMBR-seq+

### Overview

EMBR-seq libraries were prepared according to the protocol described in Wangsanuwat et al. (25). As input material for all libraries, 100 ng of total RNA was used. For the libraries denoted "EMBR-seq" and "EMBR-seq+," blocking primers were added during the polyadenylation step and reverse transcription step. For the libraries denoted "No depletion" and "RNase H," water was added instead of blocking primers during these two

steps. In the libraries denoted "RNase H" and "EMBR-seq+," the RNase H treatment was performed as described below. In the libraries denoted "No depletion" and "EMBR-seq," the RNase H treatment was skipped; the amplified RNA product of *in vitro* transcription was used directly for the library reverse transcription step as described in Wangsanuwat et al. (25).

### Blocking primer design for G. metallireducens and F. succinogenes strain UWB7

All the primers used in this study are listed in Table S2. The 16S, 23S, and 5S rRNA sequences were downloaded for *G. metallireducens GS-15* (NCBI database ASM1292v1) and *F. succinogenes* strain UWB7 (JGI IMG database Taxon ID 2703719236). Primers were designed according to the following specifications: (i) reverse complement to the reference sequences at the 3′-end of each rRNA subunit, (ii) initial primer length of approximately 30 nucleotides, and (iii) initial primer length modified to ensure a $T_m$ greater than 65°C.

Initial EMBR-seq libraries were prepared with a 50 µM blocking primer mix consisting only of the three 3′-end primers (16S, 23S, and 5S). Based on the preliminary sequencing results from these libraries, the following approach was used to determine the three to four most frequently sequenced internal rRNA genomic locations for which hotspot primers were designed. First, the reads mapping to rRNA were visualized as a cumulative distribution plot and the top 50 genomic mapping coordinates with the largest step increase in the cumulative distribution were identified. Next, these 50 locations were sorted from 5′- to 3′-end and clustered into groups based on natural breaks in the genomic coordinates. Finally, the sum of mapped reads per cluster was computed and the clusters were sorted based on this sum from highest to lowest. The top clusters with the greatest contribution to rRNA sequencing reads were identified, and the coordinates farthest downstream (towards the 3′-end) within each cluster were selected as hotspot targets. Typically, three to four hotspot targets were selected per rRNA subunit, as this offered a balance between efficient rRNA depletion and ease of primer design.

Hotspot primers were designed to bind the selected locations for additional rRNA depletion. Hotspot primers were designed according to the following specifications: (i) reverse complement to the reference sequence for each rRNA subunit, (ii) located approximately 100 nucleotides downstream (towards the 3′-end) from the start of the hotspot sequence, (iii) initial primer length of approximately 30 nucleotides, and (iv) initial primer length modified and/or initial primer position shifted upstream or downstream by a few bases to reach a $T_m$ greater than 65°C.

To ensure that the rRNA depletion probes do not result in non-specific binding, the probe sequences were aligned against the transcriptome. BlastN was used to align EMBR-seq blocking primers to each gene of the transcriptome. To minimize non-specific binding, primers were generally designed to have a BlastN *E*-value less than 1 and a sequence similarity less than 15 nucleotides of consecutive matches, following the results of a previous study (21).

### RNase H probe design

RNase H probes bind to aRNA, the IVT product, which has a sequence complementary to the original template transcriptome. Therefore, RNase H probes are designed to have the same sequence as the rRNA references. The locations of the RNase H probes were selected in the neighborhood of the 3′-end and hotspot blocking primers with a length of 50 nucleotides. The RNase H probe design can also be easily modified to target any additional abundant fragments found in prior sequencing libraries.

### Preparation of blocking primer mix

Blocking primers for the species of interest (Table S2) were individually resuspended to 100 µM. The primers were mixed such that the three 3′-end primers together were 50 µM

in the final primer mix, and all the hotspot primers together were 50 µM in the final primer mix, for a total primer mix concentration of 100 µM.

## Polyadenylation

Two microliters of total RNA (50 ng/µL) was mixed with 1 µL of 5× first-strand buffer [250 mM Tris-HCl (pH 8.3), 375 mM KCl, 15 mM MgCl$_2$, comes with Superscript II Reverse Transcriptase, Invitrogen Cat. # 18064014], 1 µL of blocking primer mix, 0.1 µL 10 mM ATP, and 0.1 µL *E. coli* poly-A polymerase (New England Biolabs, Cat. # M0276S). The samples were incubated at 37°C for 10 min. In the "No depletion" and "RNase H" libraries, 1 µL of nuclease-free water was added instead of the blocking primer mix.

## Reverse transcription

The polyadenylation product was mixed with 0.5 µL 10 mM dNTPs (New England Biolabs, Cat. # N0447L), 1 µL reverse transcription primers (25 ng/µL; Table S2), and 1.3 µL blocking primer mix and heated to 65°C for 5 min and 58°C for 1 min and then quenched on ice. In the "No depletion" and "RNase H" libraries, 1.3 µL of nuclease-free water was added instead of the blocking primer mix. This product was then mixed with 1.2 µL 5× first strand buffer, 1 µL 0.1 M DTT, 0.5 µL RNaseOUT (Thermo Fisher Scientific, Cat. # 10777019), and 0.5 µL Superscript II Reverse Transcriptase and then incubated at 42°C for 1 h. Immediately afterward, the samples were heat inactivated at 70°C for 10 min.

## Second-strand synthesis

The reverse transcription product was mixed with 33.5 µL nuclease-free water, 12 µL 5× second-strand buffer [100 mM Tris-HCl (pH 6.9), 23 mM MgCl$_2$, 450 mM KCl, 0.75 mM β-NAD, 50 mM (NH$_4$)$_2$SO$_4$; Invitrogen, Cat. # 10812014], 1.2 µL 10 mM dNTPs, 0.4 µL *E. coli* ligase (Invitrogen, Cat. # 18052019), 1.5 µL DNA polymerase I (Invitrogen, Cat. # 18010025), and 0.4 µL RNase H (Invitrogen, Cat. # 18021071) and incubated at 16°C for 2 h. The cDNA was purified with 1× AMPure XP DNA beads (Beckman Coulter, Cat. # A63881) and eluted in 24 µL nuclease-free water that was subsequently concentrated to 6.4 µL.

## In vitro transcription

The product from the previous step was mixed with 9.6 µL of *in vitro* transcription mix (1.6 µL of each ribonucleotide, 1.6 µL 10× T7 reaction buffer, 1.6 µL T7 enzyme mix; MEGAscript T7 Transcription Kit, Thermo Fisher Scientific, Cat. # AMB13345) and incubated at 37°C for 13 h. The IVT product was mixed with 6 µL ExoSAP-IT PCR Product Cleanup Reagent (Thermo Fisher Scientific, Cat. # 78200.200.UL) and incubated at 37°C for 15 min. Next, it was treated with 5.5 µL fragmentation buffer [200 mM Tris-acetate (pH 8.1), 500 mM KOAc, 150 mM MgOAc] at 94°C for 3 min and immediately quenched with 2.75 µL stop buffer (0.5 M EDTA) on ice. The fragmented aRNA was size selected with 0.8× AMPure RNA beads (RNAClean XP Kit, Beckman Coulter, Cat. # A63987) and eluted in 18 µL nuclease-free water. In the libraries denoted "No depletion" and "EMBR-seq," the fragmented and cleaned-up *in vitro* transcription product was used directly for Illumina library preparation by reverse transcription and PCR, as described previously (25, 55). In the "RNase H" and "EMBR-seq+" libraries, the RNase H treatment was performed before proceeding to Illumina library preparation.

## RNase H probe mix and buffer preparation

RNase H probes (Table S2) were individually resuspended to 100 µM and mixed equimolarly, and the mixture was diluted to 10 µM (~155 ng/µL). RNase H buffer mix was prepared by mixing 10 µL of 5× first-strand buffer with 2.7 µL of 5× second strand buffer or scaling up as needed.

## RNase H treatment for monocultures

After ExoSAP treatment, RNA fragmentation, and bead-based size selection, the aRNA concentration was measured on a NanoDrop One spectrophotometer. One thousand nanograms of aRNA was mixed with 500 ng (3.2 µL) of RNase H probe mix, 3.2 µL of RNase H buffer mix, and nuclease-free water to reach a total volume of 15 µL for each reaction. This mixture was preheated at 65°C for 5 min and quenched on ice. One microliter of RNase H (Invitrogen, Cat. # 18021071) was added, and the reaction was incubated at 16°C for 30 min. For tests with the thermostable RNase H, 1 µL of Hybridase Thermostable RNase H at 45°C [Lucigen (now Biosearch Technologies), Cat. # H39500] was added and the reaction was incubated at 45°C for 30 min. Finally, the RNase H product was size selected with 1× AMPure RNA beads and eluted in 15 µL nuclease-free water and then concentrated to 5 µL volume for Illumina library preparation.

## Enrichment of *F. succinogenes* strain UWB7 RNA from co-cultures

### Fungal mRNA removal

Total RNA extracted from co-culture pellets was concentrated to 30 µL (40–60 ng/µL) and treated with Dynabeads Oligo(dT)$_{25}$ (Invitrogen Cat # 61002), according to the manufacturer's protocol. Briefly, each sample was mixed with 30 µL of "Binding Buffer," incubated at 65°C for 2 min, chilled on ice, mixed with 30 µL of washed oligo(dT)$_{25}$ beads, and mixed at 300 rpm on a thermomixer at room temperature for 15 min. To prevent the beads from settling in the thermomixer, the samples were mixed by pipetting every 5 min. The reaction tubes were placed on a magnetic stand, and the supernatants (90–95 µL) were transferred to new tubes. The supernatants were purified with 1.8× AMPure RNA beads and eluted in 30 µL of nuclease-free water. This remaining RNA was used for preparing the "Poly-A depleted (EMBR-seq)" libraries.

### Determination of fungal rRNA sequences for RNase H depletion

To determine potential rRNA sequences for *C. churrovis*, FastQC was used to identify the most overrepresented sequences in the "Poly-A depleted (EMBR-seq)" sample, and the top hits were aligned with BLAST against the full *C. churrovis* genome. The BLAST results suggested that the most overrepresented reads were aligned on scaffold 672.

Next, all the RNA-seq reads from the "Poly-A depleted (EMBR-seq)" libraries were mapped against scaffold 672 of the *C. churrovis* genome with the Burrows-Wheeler Aligner software package (BWA) (65). This scaffold contained four repeats of three hotspots which appeared to correspond to rRNA subunits, based on the spacing between hotspots (Fig. S5). The four repeats were observed because non-unique mapping was allowed. To produce Fig. 3C, the mapping was repeated on a version of scaffold 672 that is truncated to only the first 10,000 bases (one repeat of the predicted rRNA pattern). RNase H probes were designed according to the locations of the predicted hotspots for the 3′-ends of the 28S, 18S, and 5.8S rRNA subunits in *C. churrovis*. For RNase H treatment in co-culture EMBR-seq+ libraries, the *C. churrovis* probes were applied to co-cultures with *A. robustus* as well. Although they are different fungal species, we verified that the same RNase H target sequences are highly abundant in both *A. robustus* and *C. churrovis* genomes, and the 3′-ends of their rRNA are similar enough such that fungal rRNA depletion was efficient in both co-cultures (Fig. S4 and S5).

### RNase H reaction for co-cultures

For co-culture samples, the RNase H probe mix was altered to include probes for depleting fungal rRNA along with bacterial rRNA. One thousand nanograms of aRNA derived from co-culture samples was mixed with 850 ng (5.5 µL) of co-culture RNase H probe mix, which consisted of the following: 100 ng each of the two fungal 28S probes with 50 ng each of the two fungal 18S, fungal 5.8S, and 10 bacterial RNase H probes. The aRNA and probes were mixed with 3.2 µL RNase H buffer mix and nuclease-free water

to reach a total volume of 15 µL for each reaction. The reactions were preheated at 65°C for 5 min and quenched on ice. One microliter of RNase H was added, and the reaction was incubated at 16°C for 30 min. The RNase H product was size selected with 1× AMPure RNA beads and eluted in 15 µL nuclease-free water and then concentrated to 5 µL for Illumina library preparation.

## RNA sequencing and bioinformatics analysis

Paired-end sequencing of all libraries was performed on an Illumina NextSeq 500 or HiSeq 4000 platform. All sequencing data have been deposited to Gene Expression Omnibus under the accession number GSE223404. In the sequencing libraries, read 1 contains the sample barcode (Table S2). Read 2 is mapped to the bacterial or fungal transcriptome. Prior to mapping, read 1 was trimmed to 25 bases, read 2 was trimmed to 51 bases, and only reads containing valid sample barcodes on read 1 were retained. Next, the reads were mapped to the appropriate reference transcriptome using Burrows-Wheeler Aligner (BWA) version 0.7.15 with default parameters. The "CDS from genomic FASTA" reference files for *Escherichia coli K12* substr. *MG1655* (ASM584v2 cds) and *Geobacter metallireducens GS-15* (ASM1292v1 cds) were obtained from the NCBI database. The reference transcriptome for *Fibrobacter succinogenes* strain UWB7 (Taxon ID 2703719236) was obtained from the JGI IMG database (66). The reference genomes for *Anaeromyces robustus* and *Caecomyces churrovis* were obtained from the JGI MycoCosm portal (67). Each bacterial reference transcriptome was modified by appending the 16S, 23S, and 5S rRNA sequences. To determine rRNA content for each sample, the number of reads mapped to the rRNA was taken as a fraction of total reads mapped to the bacteria.

To compare the results of this work to those of Swift et al., raw sequencing reads (fastq files) were retrieved from the Sequencing Read Archive (SRA) for NCBI BioProject PRJNA666900 as specified in Swift et al. (37). Reads 1 and 2 were trimmed to 51 bases and mapped to the reference transcriptomes for *F. succinogenes* strain UWB7, *A. robustus*, and *C. churrovis* using the same BWA pipeline described above.

CAZyme annotations were predicted for the *F. succinogenes* strain UWB7 transcriptome using dbCAN2 (43). Three tools were run: (i) HMMER: dbCAN, (ii) DIAMOND: CAZy, and (iii) HMMR: dbCAN_sub, and CAZyme annotations were accepted if they were predicted with two out of the three tools.

Differential gene expression analysis was performed using DEseq2 version 1.36.0 (68), with a minimum absolute $\log_2$(fold change) of 0.8, statistical significance threshold for an adjusted $P$ value of less than 0.05, and a minimum gene expression level of two reads per million.

To account for biases introduced by EMBR-seq+ in the *F. succinogenes* strain UWB7 monocultures (Fig. 2D), genes determined to be differentially expressed between the "No depletion" and "EMBR-seq" data set were filtered out and excluded from all analyses between monoculture and co-culture (Fig. 3E and F; Fig. S7 to S9; Tables S4 to S7). DE-Seq2 thresholds for the bias determination were $|\log_2 (\text{fold change})| > 1.5$, $P_{adj} < 0.05$, and RPM > 2. Filtering results are included in Table S4.

## ACKNOWLEDGMENTS

We would like to thank members of the Dey and O'Malley groups and Dr. Elizabeth Wilbanks for helpful discussions. The authors thank Dr. Jennifer Smith for the assistance with sequencing Illumina libraries. Sequencing was performed at the Biological Nanostructures Laboratory within the California NanoSystems Institute (CNSI), supported by the University of California, Santa Barbara (UCSB), and the University of California, Office of the President (UCOP). Additional sequencing was performed at Novogene. We further acknowledge the Center of Scientific Computing at CNSI for computational facilities that are funded by NSF MRSEC (DMR 2308708) and NSF CNS-1725797.

This work was supported by the Institute for Collaborative Biotechnologies grants W911NF-09–D-0001 and W911NF-19-2-0026 (M.A.O.), US Army Research Office contract W911NF-19-1-0010 (M.A.O.), the National Science Foundation grant MCB-1553721

(M.A.O.), NIH grants R01HD099517 and R01HG011013 (S.S.D.), the UCSB Academic Senate Faculty Research Grant (S.S.D.), and the CNSI Challenge Grant Program, supported by UCSB and UCOP (S.S.D. and M.A.O.).

K.A.H., C.W., and S.S.D. conceived the method and designed the experiments. K.A.H., C.W., and S.C.T. participated in data collection for EMBR-seq+. L.V.B. cultivated and performed RNA extractions for all monocultures and co-cultures of F. succinogenes strain UWB7, A. robustus, and C. churrovis. A.R.R. cultivated G. metallireducens. K.A.H. and C.W. analyzed the data. K.A.H., L.V.B., A.R.R., M.A.O., and S.S.D. wrote the manuscript. S.S.D. guided experimental design and data analysis. All authors have read and approved the final version of the manuscript.

## AUTHOR AFFILIATIONS

[1]Department of Chemical Engineering, University of California Santa Barbara, Santa Barbara, California, USA
[2]Biological Engineering Program, University of California Santa Barbara, Santa Barbara, California, USA
[3]Biological Sciences, University of Cincinnati, Cincinnati, Ohio, USA
[4]Neuroscience Research Institute, University of California Santa Barbara, Santa Barbara, California, USA

## PRESENT ADDRESS

Chatarin Wangsanuwat, Moderna, Inc, Cambridge, Massachusetts, USA

## AUTHOR ORCIDs

Kellie A. Heom  http://orcid.org/0009-0004-4962-9886
Michelle A. O'Malley  http://orcid.org/0000-0002-6065-8491

## FUNDING

| Funder | Grant(s) | Author(s) |
| --- | --- | --- |
| UC | UCSB | Institute for Collaborative Biotechnologies (ICB) | W911NF-09-D-0001, W911NF-19-2-0026 | Michelle A. O'Malley |
| DOD | USA | AFC | CCDC | Army Research Office (ARO) | W911NF-19-1-0010 | Michelle A. O'Malley |
| National Science Foundation (NSF) | MCB-1553721 | Michelle A. O'Malley |
| UCSB Academic Senate Faculty Research Grant | | Siddharth S. Dey |
| CNSI Challenge Grant | | Siddharth S. Dey |

## DATA AVAILABILITY

RNA-seq data are available in Gene Expression Omnibus under accession no. GSE223404.

## ADDITIONAL FILES

The following material is available online.

### Supplemental Material

**Supplemental Information (mSystems00281-23-s0001.pdf).** Figures S1–S9; Tables S1–S3, S6, and S7; captions to Tables S4 and S5.
**Table S4 (mSystems00281-23-s0002.xlsx).** Results of differential gene expression analysis.
**Table S5 (mSystems00281-23-s0003.xlsx).** Select bacterial stress response genes upregulated in co-culture.

Open Peer Review

**PEER REVIEW HISTORY (review-history.pdf).** An accounting of the reviewer comments and feedback.

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
