## [Reviewer comments · mSystems]

Targeted rRNA depletion enables efficient mRNA sequencing in diverse bacterial species and complex co-cultures

Kellie Heom, Chatarin Wangsanuwat, Lazarina Butkovich, Scott Tam, Annette Rowe, Michelle O'Malley, and Siddharth Dey

Corresponding Author(s): Siddharth Dey, University of California Santa Barbara

Review Timeline:

Submission Date:	March 23, 2023
Editorial Decision:	June 1, 2023
Revision Received:	August 15, 2023
Accepted:	September 12, 2023

Editor: Danielle Tullman-Ercek

Reviewer(s): Disclosure of reviewer identity is with reference to reviewer comments included in decision letter(s). The following individuals involved in review of your submission have agreed to reveal their identity: Christina Homberger (Reviewer #3)

Transaction Report:

DOI: <https://doi.org/10.1128/msystems.00281-23>

June 1, 2023

Prof. Siddharth Subhas Dey
University of California Santa Barbara
Chemical Engineering and Biological Engineering
Santa Barbara, CA 93106

Re: mSystems00281-23 (Targeted rRNA depletion enables efficient mRNA sequencing in diverse bacterial species and complex co-cultures)

Dear Prof. Siddharth Subhas Dey:

Thank you for submitting your manuscript to mSystems. We have completed our review and I am pleased to inform you that, in principle, we expect to accept it for publication in mSystems. However, acceptance will not be final until you have adequately addressed the reviewer comments below. In addition, I have an editorial request that you reduce the number of supplementary files to less than 10; I believe each worksheet in the Excel workbook counts as a separate file. To reduce, you may consider embedding Tables S2, S3, and S6 directly into the main supplement, for example, and perhaps rework S5AB such that they can be embedded, as well. I am available to consider ideas for how to best do this while conveying the information clearly, should you have questions about this requirement.

Preparing Revision Guidelines

Please return the manuscript within 60 days; if you cannot complete the modification within this time period, please contact me. If you do not wish to modify the manuscript and prefer to submit it to another journal, please notify me of your decision immediately so that the manuscript may be formally withdrawn from consideration by mSystems.

Sincerely,

Danielle Tullman-Ercek

Editor, mSystems

Journals Department
Reviewer comments:

Reviewer #1 (Comments for the Author):

In this study, Heom et al present EMBR+seq, which is built on the previously published EMBR-seq protocol from the same group. EMBR+seq is a total RNA seq protocol for prokaryotes with a very efficient rRNA depletion step which is very easy to extend to non-model and not-so-well annotated species. The smart trick of the protocol relies on a RNase H-based digestion step on the aRNA, which makes it very easy for further optimization of the required probes without the need of having to repeat the full experiment (including library prep). In addition, the probes target the 3' end of the rRNA, allowing for an easy removal of short rRNA left-over product on the library by AMPure XP bead selection.

The paper is very well written, and figures and data analysis are very well presented.

I have a few questions that I hope the authors can address.

- How do the authors define "hotspot sequences"? When looking at the no depletion cumulative % vs rRNA region curves, there are some spots that one would identify as hotspots that are not indicated. Would the inclusion of such regions improve the outcome of the rRNA depletion step even more? Is there a preferred number of probes to be used?

- In the co-culture experiment, after removal of the fungal mRNA using magnetic beads and removal of fungal rRNA with the scaffold-mapping-based approach, still ~40% of the reads are assigned to fungi. What are those?

- Again in the context of co-culture experiments, would EMBR+seq be a good tool to estimate fungal-bacterial ratios when those are unknown?

- Can EMBR+seq be applied to eukaryotes (e.g. human, mouse) and does it have the potential to become a single-cell-based protocol?

Reviewer #2 (Comments for the Author):

Summary: This study is an extension of Enriched mRNA by Blocked rRNA, or EMBR-Seq technology for the targeted depletion of ribosomal RNA transcripts in microbial total RNA pools. In which an additional RNase H treatment step is introduced to deplete rRNA transcripts that escaped the selective reverse transcription process. The rRNA depletion efficiency is comparable to those of existing methods (although authors did not provide relevant experimental data to allow direct comparison, and to avoid batch effect). The use of fewer oligonucleotides compared to other RNase H digestion methods is noteworthy and proves to be quite handy for RNA-Seq of mix-culture samples. Please consider the comments below:

Comments:

1) Reading through this manuscript, one cannot help but wonder the motivation of EMBR-Seq+? The rRNA depletion efficiency in its original version EMBR-Seq seems more than sufficient for a general-purpose RNA-sequencing (Fig. 1C). The only notable 'ingenuity' in this paper lies in the improved genome coverage, but the underlying rationale (i.e. technical necessity for certain biological discoveries) has not been explicitly stated in the paper. The descriptions of bacterial-fungal co-culture are mostly generalized observations (Line 417-433) and lack 'novel finding' that can give credit to EMBR-Seq+.

2) Sequencing depth is essential for the detection of genes with low basal expression (PMID: 23270466). Would be beneficial if authors explicitly state the sequencing depth for the data in Fig. 1D.

3) It seems that important statistical analysis is missing in few datasets, for example, but not limited to, Fig. 3D.

4) How many genes are filtered according to the 'threshold for the bias determination'? (in Line 716-722; supplementary Table S4). Also, what is the Pearson correlation (r^2) between the 'unfiltered' set of 'No depletion' and 'EMBR-Seq+' dataset?

5) Would benefit from pairwise comparison of gene expression levels between EMBR-Seq and EMBR-Seq+. This way, one can tell potential 'biases' arising from different probe usage and RNase H treatment.

6) In performing RNA-Seq for bacterial-fungal co-culture samples, authors enriched bacterial portion of mRNA by technically

depleting the fungal portions of mRNA. Would this give rise to biased results (Line 246-248)?

7) How does one ensure non-specific binding of rRNA probes on other parts of the genome?

8) The manuscript would benefit from providing a step-by-step, user-friendly protocol, preferably in a separate supplementary pdf, for the potential users/readers of the manuscript.

Reviewer #3 (Comments for the Author):

In their study 'Targeted rRNA depletion enables efficient mRNA sequencing in diverse bacterial species and complex co-cultures, the authors report an updated RNA-seq method (EMBR-seq+) including efficient ribosomal RNA depletion in bacteria as well as co-cultures. Their method relies on an optimized rRNA depletion protocol that combines rRNA blocking primers with RNase H-based depletion of rRNA transcripts.

The manuscript by Heom et al. is technically oriented and focuses on the optimization as well as validation of an existing bacterial RNA-seq protocol including rRNA depletion using specific blocking primers. In addition, the updated protocol was expanded by a second depletion step mediated by RNase H. For the validation of EMBR-seq+, the authors use *E. coli*, *F. succinogenes* as well as *G. metallireducens*. Using the improved EMBR-seq+ protocol, the rRNA depletion efficiency was further improved especially in co-cultures of bacteria and anaerobic fungi. By applying EMBR-seq+ on a co-culture of *F. succinogenes* and *A. robustus*, the authors were able to gain insights into the change of CAZyme expression profiles in the absence and presence of the anaerobic fungi *A. robustus*.

Overall, the method presented by Heom et al. would be of high interest in the field and represents a useful addition to their published EMBR-seq protocol. The optimized rRNA depletion protocol is especially of high interest for transcriptomics studies of the gut microbiome or any other non-model organisms where current rRNA depletion approaches are highly limited in their efficiencies. Another important application for EMBR-seq+ is the ability to efficiently deplete rRNA even in a dual-RNAseq approach covering bacterial as well as fungal RNA.

In general, EMBR-seq+ represents a valuable addition to the field of bacterial transcriptomics and therefore, I highly recommend this manuscript for publication.

Major points:

- The authors should generally point out in more detail where EMBR-seq+ has its main advantages over their previously published EMBR-seq protocol. Especially as EMBR-seq already uses only a few oligonucleotides per rRNA, is highly cost-efficient and shows a high rRNA depletion efficiency. According to the data presented, the main advantage of EMBR-seq+ is its high depletion efficiency in co-cultures and complex samples, not so much for monocultures. For the reader, this would be helpful to be already mentioned in the abstract and/or introduction. In addition, a direct comparison of both approaches in Supplementary Table S1 is also recommended.

- Lines 394-397: The statement that bacterial single-cell RNA-seq protocols do not include any rRNA depletion is no longer current. There are several methods published recently including the removal of unwanted ribosomal RNA or the use of probes designed to capture only mRNA transcripts:

PMIDs: 36708705; 36880749; 37012420; <https://www.biorxiv.org/content/10.1101/2022.09.21.508688v2>

This should be corrected accordingly and supplemented with these additional references.

Besides, not all mentioned scRNA-seq methods include an IVT amplification step, thereby preventing the direct application of EMBR-seq+. This requires further clarification in the text as well.

- Regarding the potential application of EMBR-seq+ on low-input or even single-cell samples, the authors should comment on the minimal input required for EMBR-seq+ to enable an efficient rRNA depletion while obtaining sufficient yield for sequencing.

- Figure 3F, Supplementary Figure S8: The DESeq2 analysis performed in the monoculture of *F. succinogenes* shows reproducible data across all replicates, whereas the replicates for the co-culture show clear differences in the gene expression profiles. The authors should comment on this.

- Supplementary Figure S9: The DESeq2 data for *C. churrovii* shows clear differences among replicates of monoculture as well as co-culture. Also at this point, the authors should comment on the discrepancies.

Minor points:

- For the investigated CAZyme, one or two descriptive sentences are required in the introduction. Also, the abbreviation CAZyme is not defined.

- Lines 77-83: These statements require appropriate references.

- Lines 91-94: Supporting references are missing.

- Line 269: to quantify.

- Line 386: A major change that we introduced in EMBR-seq+ compared to EMBR-seq is the inclusion...

- Lines 375-379, 434-436, and 439-440: These sections are highly redundant and can be shortened.

- Spelling of the species names in the references should be harmonized (only some are written in italics, e.g. lines 769, 808).

- Figure 3B and S4: The different conditions shown in this figure should also be indicated with the terms EMBR-seq and EMBR-seq+ to stay consistent. This also includes some clarification in the corresponding text (lines 239-248).

- The replicate numbers shown for the DESeq2 analysis should be kept in order for all shown experiments (e.g. Figure S8, S9).

Supplementary Information:

- Line 62: libraries.

Manuscript title: Targeted rRNA depletion enables efficient mRNA sequencing in diverse bacterial species and complex co-cultures

Summary: This study is an extension of Enriched mRNA by Blocked rRNA, or EMBR-Seq technology for the targeted depletion of ribosomal RNA transcripts in microbial total RNA pools. In which additional RNase H treatment step is introduced to deplete rRNA transcripts that escaped the selective reverse transcription process. The rRNA depletion efficiency is comparable to those of existing methods (although authors did not provide relevant experimental data to allow direct comparison, and to avoid batch effect). The use of fewer oligonucleotides in comparison to other RNase H digestion method are noteworthy, and proves to be quite handy for RNA-Seq of mix-culture samples. Please consider the comments below:

Comments:

- 1) Reading through this manuscript, one cannot help but wonder the motivation of EMBR-Seq+. The rRNA depletion efficiency in its original version EMBR-Seq seems more than sufficient for a general-purpose RNA-sequencing (Fig. 1C). The only notable 'ingenuity' in this paper lies in the improved genome coverage, but the underlying rationale (i.e. technical necessity for certain biological discoveries) has not been explicitly stated in the paper. The descriptions of bacterial-fungal co-culture are mostly generalized observations (Line 417-433) and lack 'novel finding' that can give credit to EMBR-Seq+.
- 2) Sequencing depth is essential for the detection of genes with low basal expression (PMID: 23270466). Would be beneficial if authors explicitly state the sequencing depth for the data in Fig. 1D.
- 3) It seems that important statistical analysis is missing in few datasets, for example, but not limited to, Fig. 3D.
- 4) How many genes are filtered according to the 'threshold for the bias determination?' (in Line 716-722; supplementary Table S4). Also, what is the Pearson correlation (r^2) between the 'unfiltered' set of 'No depletion' and 'EMBR-Seq+' dataset?
- 5) Would benefit from pairwise comparison of gene expression levels between EMBR-Seq and EMBR-Seq+. This way, one can tell potential 'biases' arising from different probe usage and RNase H treatment.
- 6) In performing RNA-Seq for bacterial-fungal co-culture samples, authors enriched bacterial portion of mRNA by technically depleting the fungal portions of mRNA. Would this give rise to biased results (Line 246-248)?
- 7) How does one ensure non-specific binding of rRNA probes on other parts of the genome?
- 8) The manuscript would benefit from providing a step-by-step, user-friendly protocol, preferably in a separate supplementary pdf, for the potential users/readers of the manuscript.

Response to Reviewer #1

We thank Reviewer #1 for their insightful questions, which we have addressed in the revised manuscript. Remarks of Reviewer #1 are denoted in *italics*.

In this study, Heom et al present EMBR+seq, which is built on the previously published EMBR-seq protocol from the same group. EMBR+seq is a total RNA seq protocol for prokaryotes with a very efficient rRNA depletion step which is very easy to extend to non-model and not-so-well annotated species. The smart trick of the protocol relies on a RNase H-based digestion step on the rRNA, which makes it very easy for further optimization of the required probes without the need of having to repeat the full experiment (including library prep). In addition, the probes target the 3' end of the rRNA, allowing for an easy removal of short rRNA left-over product on the library by AMPure XP bead selection.

The paper is very well written, and figures and data analysis are very well presented.

We thank the reviewer for finding our manuscript well-written and well-presented, and for finding the RNase H-based digestion very useful for rapid optimization of probe design without having to repeat the full experiment. We have addressed the comments of Reviewer #1 in detail below:

I have a few questions that I hope the authors can address.

1) How do the authors define "hotspot sequences"? When looking at the no depletion cumulative % vs rRNA region curves, there are some spots that one would identify as hotspots that are not indicated. Would the inclusion of such regions improve the outcome of the rRNA depletion step even more? Is there a preferred number of probes to be used?

The hotspot sequences for each species were determined based on preliminary sequencing results, where EMBR-seq was performed with only three blocking primers; one each at the 3'-ends of the 16S, 23S and 5S rRNA subunits. From the sequencing results of EMBR-seq without hotspot primers, reads mapping to rRNA were visualized as a cumulative distribution plot and 3-4 "hotspots" were systematically selected using the approach described below. As an example, we show the preliminary data for *F. succinogenes* strain UWB7 16S rRNA subunit and outline how we selected the final hotspot targets based on the most frequent genomic mapping coordinates: 670, 871, and 964 (Figure R1).

- a. Generate a cumulative distribution of mapped sequencing reads along an rRNA subunit and identify the top 50 genomic coordinates with the largest step increase in the cumulative distribution.
- b. Sort the top 50 locations from the 5'- to '3'-end and cluster them into groups based on natural breaks in the genomic coordinates.
- c. Compute the sum of mapped reads for each cluster and sort the clusters based on this sum from highest to lowest.
- d. Design hotspot primers corresponding to the top clusters until the desired percentage of reads are covered. As an example, for the "EMBR-seq" 16S rRNA data shown in Figure R1, the top 3 clusters covered 26%, 25%, and 17% of the mapped reads (within the subset of reads in the top 50 genomic coordinates). Within each cluster, the coordinate that is farthest downstream (towards the 3'-end) was selected as the hotspot target. For designing primers, we selected only the top 3 hotspots to limit the size of the primer set.

Figure R1. Cumulative percentage of reads mapping along *F. succinogenes* strain UWB7 16S rRNA for “No depletion” and “EMBR-seq” libraries with only 3'-end blocking primers.

To make it easier for other users of this technology to design hotspot primers, we have now included these details in the Methods section of the revised manuscript (lines 591-603):

“Based on the preliminary sequencing results from these libraries, the following approach was used to determine the 3-4 most frequently sequenced internal rRNA genomic locations for which hotspot primers were designed. First, the reads mapping to rRNA were visualized as a cumulative distribution plot and the top 50 genomic mapping coordinates with the largest step increase in the cumulative distribution were identified. Next, these 50 locations were sorted from 5'- to 3'-end and clustered into groups based on natural breaks in the genomic coordinates. Finally, the sum of mapped reads per cluster was computed and the clusters were sorted based on this sum from highest to lowest. The top clusters with the greatest contribution to rRNA sequencing reads were identified and the coordinates farthest downstream (towards the 3'-end) within each cluster were selected as hotspot targets. Typically, 3-4 hotspot targets were selected per rRNA subunit, as this offered a balance between efficient rRNA depletion and ease of primer design.”

Finally, we agree with the reviewer that while increasing the number of hotspot probes increases rRNA depletion, the gain in rRNA depletion efficiency diminishes as more probes are included. Further, increasing the number of hotspot probes also increases the chances of off-target depletion. Therefore, to strike a balance between the number of hotspot probes included/ease of design and rRNA depletion efficiency, we find that 4-5 probes (hotspot and 3'-end probes) per 16S and 23S rRNA and one probe per 5S rRNA (~9-10 probes per microbial species) generally works well.

2) *In the co-culture experiment, after removal of the fungal mRNA using magnetic beads and removal of fungal rRNA with the scaffold-mapping-based approach, still ~40% of the reads are assigned to fungi. What are those?*

We think these reads mapping to the fungal genome arise due to the following reasons: (1) As the fungal genomes are not well-annotated, we used the scaffold-mapping-based approach to identify the putative rRNA sequences, and designed RNase H probes against them at the 3'-end. It is possible that this lack of annotation resulted in less efficient rRNA depletion. (2) While we used poly-T bead-based pull down to deplete fungal mRNA, it is likely that a small fraction of mRNA was not captured, resulting in the sequencing of these molecules downstream. (3) Another consideration is that we mapped reads to the entire fungal genome, rather than the coding transcriptome as these genomes are not well-annotated resulting in transcriptome references that are likely incomplete. However, a consequence of this, together with the poly-adenylation step in EMBR-seq+, is that we potentially also sequence and map non-polyadenylated non-coding RNA. In the revised manuscript, we have now included the following text to discuss sources that give rise to reads mapping to the fungal genome (lines 286-298):

“We hypothesize that we still obtained reads mapping to the fungal genome due to the following reasons. First, the RNase H probes to deplete fungal rRNA were designed based on putative rRNA sequences resulting from our scaffold-mapping-based approach. The lack of experimentally validated rRNA annotation may have led to suboptimal RNase H probe designs and less efficient rRNA depletion. Second, our poly-T pull down may not have captured a small fraction of fungal mRNA, resulting in these molecules being sequenced. Third, we mapped our sequencing reads to the entire fungal genome, rather than the coding transcriptome as these genomes are not well-annotated resulting in transcriptome references that are likely incomplete. However, a consequence of this, together with the poly-adenylation step in EMBR-seq+, is that we potentially also sequenced and mapped non-polyadenylated non-coding RNA. Nevertheless, our combined fungal RNA depletion approach enabled us to effectively enrich bacterial RNA from our co-cultures.”

3) *Again in the context of co-culture experiments, would EMBR+seq be a good tool to estimate fungal-bacterial ratios when those are unknown?*

While EMBR-seq+ sequencing could provide an estimate of the fungal-bacterial ratios in a consortium, total RNA extraction and rRNA depletion steps could introduce potential biases in this measurement. An alternate strategy to estimate this ratio would be to perform genomic DNA sequencing. In addition to this, other potential approaches could include physical cell counting where possible, or orthogonal optical measurements such as fungal autofluorescence intensity and bacterial absorbance (Leggieri *et al.*, 2021).

4) *Can EMBR+seq be applied to eukaryotes (e.g. human, mouse) and does it have the potential to become a single-cell-based protocol?*

By depleting rRNA, EMBR-seq+ can potentially be used for total RNA sequencing in eukaryotes. In our group, we have recently generated preliminary data showing efficient rRNA depletion in the ES-E14TG2a (E14) mouse embryonic stem cell line, enabling us to quantify both mRNA and non-coding RNA at high resolution (Figure R2). In the future, we plan to use EMBR-seq+ to investigate the function of non-polyadenylated non-coding RNA in mammalian cells, as well as interactions

between mammalian cells and pathogenic microbes. As this is outside the scope of the current manuscript, we have not included these details in the revised manuscript.

Figure R2. Percent of sequencing reads mapping to rRNA, mRNA and non-coding RNA in E14 mouse embryonic stem cells. Dots indicate individual biological replicates.

In addition, we are currently working towards scaling EMBR-seq+ down to single bacterial cells and we think it is feasible to accomplish this. In the revised manuscript, we have expanded our discussion of bacterial single-cell RNA sequencing methods as follows (lines 433-446):

“Additionally, while many early bacterial single-cell RNA-seq methods lacked an rRNA depletion step, more recent reports employ CRISPR-Cas9-mediated rRNA depletion or the use of probes to only capture mRNA transcripts of interest (Blattman *et al.*, 2020; Imdahl *et al.*, 2020; Kuchina *et al.*, 2020; Homberger *et al.*, 2023; McNulty *et al.*, 2023). However, as the RNase H-based digestion strategy described in this work requires only a few probes per rRNA species, in the future it could readily be integrated with single-cell RNA-seq methods that involve IVT-based amplification (Hashimshony *et al.*, 2012, 2016; Wang *et al.*, 2022; Ma *et al.*, 2023). In addition, the use of blocking primers to deplete rRNA can potentially also be scaled down to single cells as we previously showed that EMBR-seq efficiently depletes rRNA from as little as 0.02 ng input total RNA (Wangsanuwat *et al.*, 2020). This corresponds to a pool of ~200 single bacteria (assuming ~0.1 pg of RNA per bacterial cell) and is in line with the sensitivity requirements of the existing methods that employ combinatorial barcoding strategies (Blattman *et al.*, 2020; Kuchina *et al.*, 2020; Wang *et al.*, 2022; Ma *et al.*, 2023). Therefore, we anticipate that in the future EMBR-seq+ could potentially be applied to single cells, thereby adding to the growing toolkit for profiling individual bacterial cells.”

Response to Reviewer #2

We thank Reviewer #2 for their thoughtful feedback, which has been incorporated into the revised manuscript. Remarks of Reviewer #2 are denoted in *italics*.

Summary: This study is an extension of Enriched mRNA by Blocked rRNA, or EMBR-Seq technology for the targeted depletion of ribosomal RNA transcripts in microbial total RNA pools. In which an additional RNase H treatment step is introduced to deplete rRNA transcripts that escaped the selective reverse transcription process. The rRNA depletion efficiency is comparable to those of existing methods (although authors did not provide relevant experimental data to allow direct comparison, and to avoid batch effect). The use of fewer oligonucleotides compared to other RNase H digestion methods is noteworthy and proves to be quite handy for RNA-Seq of mix-culture samples. Please consider the comments below:

We thank the reviewer for finding the use of fewer oligonucleotides for RNase H-based digestion in EMBR-seq+ compared to other methods to be noteworthy. We have addressed the comments of Reviewer #2 in detail below:

Comments:

1) Reading through this manuscript, one cannot help but wonder the motivation of EMBR-Seq+? The rRNA depletion efficiency in its original version EMBR-Seq seems more than sufficient for a general-purpose RNA-sequencing (Fig. 1C). The only notable 'ingenuity' in this paper lies in the improved genome coverage, but the underlying rationale (i.e. technical necessity for certain biological discoveries) has not been explicitly stated in the paper. The descriptions of bacterial-fungal co-culture are mostly generalized observations (Line 417-433) and lack 'novel finding' that can give credit to EMBR-Seq+.

We agree with the reviewer that while EMBR-seq is efficient, the inclusion of RNase H-based depletion of rRNA in EMBR-seq+ brings the efficiency of the method in line with commercial kits, with depletion efficiencies of up to 99% for monocultures. Importantly, for non-model microbial co-cultures, the improved efficiency of EMBR-seq+ over EMBR-seq and RNase H-based depletion alone is more clearly evident (Figure 3D). EMBR-seq+ resulted in 18.4% and 23.9% rRNA reads in the sequencing data (for *F. succinogenes* strain UWB7 co-cultured with *A. robustus* and *C. churrovis*, respectively) and outperformed both EMBR-seq (with 28.3% and 32.5% rRNA reads) and RNase H-based depletion alone (with 27.0% and 42.9% rRNA reads).

In addition to the improved genome coverage, we would like to note that EMBR-seq+ offers several key advantages: (1) Compared to other methods and commercial kits, EMBR-seq+ requires a much smaller set of probes (typically fewer than 10 per rRNA species which is approximately an order of magnitude fewer probes than other methods), to achieve similar rRNA depletion efficiency. (2) This small probe set enables EMBR-seq+ to be highly customizable and easily applicable to diverse bacterial species and reduces costs significantly compared to other methods. (3) The RNase H-based rRNA depletion on IVT amplified RNA in EMBR-seq+ allows for substantial flexibility in late-stage refinements to the list of depletion targets without having to repeat the experiment from total RNA. We anticipate that the high rRNA depletion efficiency, ease of probe design, low cost, and high flexibility of EMBR-seq+ will enable rapid and detailed profiling of the transcriptome of non-model microbial systems. Finally, we would like to note that in the future, EMBR-seq+ can potentially also be scaled down to single cells and expanded to studies for quantifying non-polyadenylated non-coding RNA in eukaryotes. For a detailed discussion on this, please see our response to Reviewer #1, comment 4.

In addition to the technical features, in this work we show that the improved genome coverage of EMBR-seq+ enabled systematic investigation of the transcriptomic remodeling of *F. succinogenes* strain UWB7 when cultured with fungi. Specifically, compared to a previous study, EMBR-seq+ was significantly more efficient at capturing bacterial mRNA within co-cultures, and we used this to demonstrate that specific CAZymes in *F. succinogenes* strain UWB7 were differentially expressed in the presence of fungi, which has not been reported previously (Supplementary Figure S3, lines 250-254) (Swift *et al.*, 2021). *F. succinogenes* has been an extensively studied cellulolytic bacterial species, because it was frequently identified in rumen samples and found to encode a large number of CAZymes in its genome (Tajima *et al.*, 2001; Krause *et al.*, 2003; Shinkai and Kobayashi, 2007; Suen *et al.*, 2011; Burnet *et al.*, 2015; Raut *et al.*, 2019; Stewart *et al.*, 2019). More recently, anaerobic gut fungal genomes have also been discovered to be exceptionally abundant with CAZymes (Solomon *et al.*, 2016; Haitjema *et al.*, 2017; Seppälä *et al.*, 2017). Our co-culture experiments demonstrate the interactions between the two species and show that *F. succinogenes* strain UWB7 downregulates several lignocellulose-degrading glycoside hydrolase genes, suggesting that in the presence of the gut fungi, sufficient sugar is liberated by the fungi such that the bacteria reprogram their transcriptional state. For a detailed discussion of the remodeling of the bacterial transcriptome in co-cultures and the novel biological results presented in this manuscript, please also see the main manuscript (Figure 3E-F and Supplementary Figures S7-9, lines 363-412).

To reinforce the advantages of EMBR-seq+, as well as to address comment 1 from Reviewer #3, we have edited the following text in the Introduction section of the revised manuscript (lines 129-138):

“In addition, we show that EMBR-seq+ can be easily applied to diverse bacterial species, and compared to EMBR-seq, EMBR-seq+ enables rapid iterations in probe design without having to restart experiments from total RNA to deplete rRNA from microbial species with poorly annotated genomes and unknown rRNA sequences. We demonstrate this in a co-culture system of anaerobic bacteria and fungi native to the rumen microbiome and show that EMBR-seq+ is instrumental in providing deeper coverage of the bacterial transcriptome. In particular, for these bacterial-fungal co-cultures, we found that the RNase H step enabled EMBR-seq+ to outperform both EMBR-seq and RNase H-based depletion alone in terms of rRNA depletion efficiency.”

Finally, to provide more context for each of the species in the co-culture experiment, we have included some additional background and references in the revised manuscript (lines 235-246):

“As an example, anaerobic organisms isolated from the rumen microbiome are particularly interesting because of their ability to grow on raw, unprocessed substrates such as grass and depolymerize lignocellulose, a feature attributed to carbohydrate-active enzymes (CAZymes) found in their genomes. *F. succinogenes* is an extensively studied cellulolytic bacterial species, because it is frequently identified in rumen samples and found to encode a large number of CAZymes in its genome (Tajima *et al.*, 2001; Krause *et al.*, 2003; Shinkai and Kobayashi, 2007; Suen *et al.*, 2011; Burnet *et al.*, 2015; Raut *et al.*, 2019; Stewart *et al.*, 2019). More recently, anaerobic gut fungal genomes have also been discovered to be exceptionally abundant with CAZymes (Solomon *et al.*, 2016; Haitjema *et al.*, 2017; Seppälä *et al.*, 2017). As these species exist in consortia in nature rather than in isolation, understanding interactions between different species is key to gaining insights into their function in native environments and for engineering stable microbial systems *in vitro* (Gilmore *et al.*, 2019; Swift *et al.*, 2019, 2021).”

2) *Sequencing depth is essential for the detection of genes with low basal expression (PMID: 23270466). Would be beneficial if authors explicitly state the sequencing depth for the data in Fig. 1D.*

We agree with the reviewer that deeper sequencing results in detecting more lowly expressed genes; therefore, when comparing across different samples, we controlled for the variability in sequencing depth. For the data presented in Figure 1D, the sequencing depth ranged from 1.3-2.6 million reads per library. Therefore, we downsampled each library to the same sequencing depth of 1 million total reads to find that increased rRNA depletion in EMBR-seq+ results in the detection of more genes above different expression thresholds.

To ensure clarity, we have now updated the captions of Figures 1 and 2, and Supplementary Figures S2 and S6 in the revised manuscript (lines 1050-1051 and 1069-1070) and revised Supplementary Information (lines 43 and 101-102) as follows:

“For consistent comparison across methods and to control for the variability in sequencing depth across samples, panels show data that has been downsampled to 1 million sequencing reads.”

3) *It seems that important statistical analysis is missing in few datasets, for example, but not limited to, Fig. 3D.*

We thank the reviewer for this comment. We have now included statistical tests for data in Figures 1C, 2A and 3D. These statistical tests comparing rRNA depletion between different methods are described in the figure legends (lines 1042-1044, 1056-1058, 1093-1099).

4) *How many genes are filtered according to the 'threshold for the bias determination? (in Line 716-722; supplementary Table S4). Also, what is the Pearson correlation (r^2) between the 'unfiltered' set of 'No depletion' and 'EMBR-Seq+' dataset?*

A total of 239 genes out of 3168 genes in *F. succinogenes* strain UWB7 were filtered out. We did this to be conservative in our quantification of transcriptional reprogramming of the bacteria between monoculture and co-culture conditions, and to minimize any technique-specific biases in our analysis. The lists of differentially expressed genes, unfiltered and filtered, are both available in Supplementary Table S4. The details are as follows: Columns P and Q (“UR_Mono” and “UR_Coc”) indicate the upregulated genes in monoculture and co-culture conditions, respectively, before filtering. Column R (“Alpha”) indicates the 239 genes that were filtered out. Finally, Column S (“Beta”) contains the results after filtering; a “1” indicates that a gene was upregulated in monoculture and a “-1” indicates that a gene was upregulated in co-culture.

The Pearson correlation between “No depletion” and “EMBR-seq+” for the unfiltered *F. succinogenes* UWB7 monoculture data is 0.94. This is also shown in Figure 2D and the corresponding figure legend.

Based on the reviewers’ questions, we have now updated the caption for Supplementary Table S4 as follows to increase clarity (lines 162-164, 166, and 167-168):

“Results of differential gene expression analysis for comparisons between *F. succinogenes* strain UWB7 grown in monoculture vs. co-culture, using “No depletion”, “EMBR-seq”, “RNase H”, and “EMBR-seq+” libraries. For EMBR-seq+ results, predicted CAZyme annotations from dbCAN2

are also included (columns U-Z where applicable). In the table, columns I-N (named “M1”...“C3”) show RPM for each gene (calculated as raw counts * 10^6 / sum of raw counts). Columns P and Q (“UR_Mono” and “UR_Coc”) indicate the upregulated genes before filtering in monoculture and co-culture conditions, respectively. Column R (named “Alpha”) indicates genes that are differentially expressed between the “No Depletion” and “EMBR-seq+” libraries, and are filtered out in column S (named “Beta”). A total of 239 genes are filtered out. In column S, a value of 0 indicates a gene that is not differentially expressed or is differentially expressed but filtered out, -1 indicates a gene that is upregulated in co-culture, and 1 indicates a gene that is upregulated in monoculture. This supplementary table is provided as a separate Excel file.”

5) Would benefit from pairwise comparison of gene expression levels between EMBR-Seq and EMBR-Seq+. This way, one can tell potential 'biases' arising from different probe usage and RNase H treatment.

We thank the reviewer for this question. The scatterplots below show the correlation of gene expression between “EMBR-seq” and “EMBR-seq+” datasets for *E. coli* (Pearson’s $r = 0.991$), *F. succinogenes* strain UWB7 (Pearson’s $r = 0.993$), and *G. metallireducens* (Pearson’s $r = 0.990$) monocultures (Figure R3). This high correlation suggests that the RNase H-based depletion in EMBR-seq+ introduces minimal biases and that the transcriptomes obtained from EMBR-seq and EMBR-seq+ are very similar.

Figure R3. Correlation of gene expression between “EMBR-seq” and “EMBR-seq+” datasets for *E. coli* (left, Pearson’s $r = 0.991$), *F. succinogenes* strain UWB7 (center, Pearson’s $r = 0.993$), and *G. metallireducens* (right, Pearson’s $r = 0.990$) monocultures.

Based on the reviewer’s suggestion, we have added the Pearson’s r values to Supplemental Table S3 (lines 144-154) and the following text to the revised manuscript (lines 200-202):

“Similarly, the transcriptomes obtained from EMBR-seq and EMBR-seq+ were highly correlated, suggesting that RNase H-based depletion in EMBR-seq+ does not introduce technical biases. (Supplementary Table S3).”

6) In performing RNA-Seq for bacterial-fungal co-culture samples, authors enriched bacterial portion of mRNA by technically depleting the fungal portions of mRNA. Would this give rise to biased results (Line 246-248)?

Depleting the fungal mRNA using poly-T magnetic beads does not introduce biases in the bacterial transcriptome as bacterial mRNA is not poly-adenylated. If researchers are interested in

profiling both the bacterial and fungal mRNA from co-cultures, we would recommend performing poly-T bead pulldown on total RNA from co-cultures and using the bead-bound fraction to quantify the fungal mRNA (for example, using CEL-seq2 (Hashimshony *et al.*, 2016)) while the supernatant fraction can be used to quantify the bacterial mRNA using EMBR-seq+.

As our goal in this manuscript was only to characterize the bacterial mRNA and not the fungal mRNA, we have clarified this in the revised manuscript as follows (lines 262-268):

“We hypothesized that this large excess of fungal reads in our dataset derived from an initial overabundance of fungal RNA in the co-culture total RNA pellets, and a lack of targeting to deplete fungal RNA in EMBR-seq+. As our goal was to characterize the transcriptome of *F. succinogenes* strain UWB7 within the co-culture at high resolution, we aimed to deplete the fungal mRNA and rRNA. We first used poly-T magnetic beads to deplete polyadenylated fungal mRNA prior to applying EMBR-seq, resulting in a library (denoted as “Poly-A depleted (EMBR-seq+)”) with 24.9% bacteria reads.”

7) How does one ensure non-specific binding of rRNA probes on other parts of the genome?

To ensure that the rRNA depletion probes do not result in non-specific binding, we aligned the probe sequences against the transcriptome. Specifically, we used BlastN to align EMBR-seq blocking primers to each gene in the transcriptome, following the protocol of a previous study that showed gene expression was affected by nonspecific primer binding activity only when the BlastN E-value was greater than 1 and the sequence similarity was longer than 15 nucleotides of consecutive matches (Choe *et al.*, 2021). For the *E. coli* blocking primer set, we found only one primer-gene pair where the E-value was greater than 1 and continuous base alignment was greater than 15 (*E. coli* primer 16S 682 & Gene AAC76588). However, for this gene as well, the expression level (RPM) in the “EMBR-seq+” dataset was not deflated compared to the “No depletion” condition ($\log_2FC = 0.24$). Further, genome-wide comparison to the “No depletion” condition also shows that EMBR-seq+ results in minimal bias in gene expression levels (Figure 1E and 2D, and Supplemental Figure S2C, S6C, and S6F). More generally, limiting the size of the primer set, as is the case with EMBR-seq+, reduces the chances of non-specific binding. We have added the details of the primer alignment analysis to the Methods section (lines 611-616):

“To ensure that the rRNA depletion probes do not result in non-specific binding, the probe sequences were aligned against the transcriptome. BlastN was used to align EMBR-seq blocking primers to each gene of the transcriptome. To minimize non-specific binding, primers were generally designed to have a BlastN E-value less than 1 and a sequence similarity less than 15 nucleotides of consecutive matches, following the results of a previous study (Choe *et al.*, 2021).”

8) The manuscript would benefit from providing a step-by-step, user-friendly protocol, preferably in a separate supplementary pdf, for the potential users/readers of the manuscript.

We would like to thank the reviewer for this suggestion. We are currently working on a separate protocols paper where we will provide a step-by-step guide to performing EMBR-seq+. In the meantime, users of EMBR-seq+ can contact Dr. Dey via email and we will share a bench top protocol and/or help troubleshoot experiments.

Response to Reviewer #3

We thank Reviewer #3 for their helpful comments, which has significantly improved the manuscript. Remarks of Reviewer #3 are denoted in italics.

In their study 'Targeted rRNA depletion enables efficient mRNA sequencing in diverse bacterial species and complex co-cultures, the authors report an updated RNA-seq method (EMBR-seq+) including efficient ribosomal RNA depletion in bacteria as well as co-cultures. Their method relies on an optimized rRNA depletion protocol that combines rRNA blocking primers with RNase H-based depletion of rRNA transcripts.

The manuscript by Heom et al. is technically oriented and focuses on the optimization as well as validation of an existing bacterial RNA-seq protocol including rRNA depletion using specific blocking primers. In addition, the updated protocol was expanded by a second depletion step mediated by RNase H. For the validation of EMBR-seq+, the authors use E. coli, F. succinogenes as well as G. metallireducens. Using the improved EMBR-seq+ protocol, the rRNA depletion efficiency was further improved especially in co-cultures of bacteria and anaerobic fungi. By applying EMBR-seq+ on a co-culture of F. succinogenes and A. robustus, the authors were able to gain insights into the change of CAZyme expression profiles in the absence and presence of the anaerobic fungi A. robustus.

Overall, the method presented by Heom et al. would be of high interest in the field and represents a useful addition to their published EMBR-seq protocol. The optimized rRNA depletion protocol is especially of high interest for transcriptomics studies of the gut microbiome or any other non-model organisms where current rRNA depletion approaches are highly limited in their efficiencies. Another important application for EMBR-seq+ is the ability to efficiently deplete rRNA even in a dual-RNAseq approach covering bacterial as well as fungal RNA.

In general, EMBR-seq+ represents a valuable addition to the field of bacterial transcriptomics and therefore, I highly recommend this manuscript for publication.

We thank the reviewer for noting that EMBR-seq+ would be of high interest to the field of bacterial transcriptomics, and especially for profiling non-model microbes. We would also like to thank the reviewer for recommending the manuscript for publication. We have addressed the reviewer's comments in detail below.

Major points:

1) The authors should generally point out in more detail where EMBR-seq+ has its main advantages over their previously published EMBR-seq protocol. Especially as EMBR-seq already uses only a few oligonucleotides per rRNA, is highly cost-efficient and shows a high rRNA depletion efficiency. According to the data presented, the main advantage of EMBR-seq+ is its high depletion efficiency in co-cultures and complex samples, not so much for monocultures. For the reader, this would be helpful to be already mentioned in the abstract and/or introduction. In addition, a direct comparison of both approaches in Supplementary Table S1 is also recommended.

We thank the reviewer for these useful comments. For details on the main advantages of EMBR-seq+, please see our response to Reviewer #2, comment 1. Briefly, the two key advantages of EMBR-seq+ over EMBR-seq are as follows: (1) The rRNA depletion efficiency of EMBR-seq+ is higher than EMBR-seq, especially for more complex co-culture systems. (2) The RNase H-based depletion of rRNA from IVT amplified RNA allows for rapid iterations in RNase H-based probe design without having to repeat the experiments from total RNA. We have accordingly updated the Introduction section of the revised manuscript as follows (lines 129-138):

“In addition, we show that EMBR-seq+ can be easily applied to diverse bacterial species, and compared to EMBR-seq, EMBR-seq+ enables rapid iterations in probe design without having to restart experiments from total RNA to deplete rRNA from microbial species with poorly annotated genomes and unknown rRNA sequences. We demonstrate this in a co-culture system of anaerobic bacteria and fungi native to the rumen microbiome and show that EMBR-seq+ is instrumental in providing deeper coverage of the bacterial transcriptome. In particular, for these bacterial-fungal co-cultures, we found that the RNase H step enabled EMBR-seq+ to outperform both EMBR-seq and RNase H-based depletion alone in terms of rRNA depletion efficiency.”

Finally, the Discussion section of the manuscript (lines 425-433) also reinforces these points.

In addition, we have updated Supplementary Table S1 to include an entry for EMBR-seq to compare it more easily with EMBR-seq+.

Supplier or publication	Working principle	Cost/sample (approx.)	Number of primers per species	Percent rRNA left	Notes
This work (EMBR-seq+)	Blocked reverse transcription & RNase H digestion	\$8.29	17-20	1-10% (monocultures), 20% (co-cultures)	
EMBR-seq	Blocked reverse transcription	\$0.39 (with hotspot BPs)	3-10	3-20% (monocultures)	Tested with and without hotspot primers

2) Lines 394-397: *The statement that bacterial single-cell RNA-seq protocols do not include any rRNA depletion is no longer current. There are several methods published recently including the removal of unwanted ribosomal RNA or the use of probes designed to capture only mRNA transcripts:PMIDs:36708705;36880749;37012420;*

<https://www.biorxiv.org/content/10.1101/2022.09.21.508688v2>

This should be corrected accordingly and supplemented with these additional references. Besides, not all mentioned scRNA-seq methods include an IVT amplification step, thereby preventing the direct application of EMBR-seq+. This requires further clarification in the text as well.

We thank the reviewer for pointing out these recent and exciting papers, which we have included in the revised manuscript. To include these emerging bacterial scRNA-seq techniques, we have revised the following text in the Discussion section (lines 433-439):

“Additionally, while many early bacterial single-cell RNA-seq methods lacked an rRNA depletion step, more recent reports employ CRISPR-Cas9-mediated rRNA depletion or the use of probes to only capture mRNA transcripts of interest (Blattman *et al.*, 2020; Imdahl *et al.*, 2020; Kuchina *et al.*, 2020; Homberger *et al.*, 2023; McNulty *et al.*, 2023). However, as the RNase H-based digestion strategy described in this work requires only a few probes per rRNA species, in the future it could readily be integrated with single-cell RNA-seq methods that involve IVT-based amplification (Hashimshony *et al.*, 2012, 2016; Wang *et al.*, 2022; Ma *et al.*, 2023).”

3) *Regarding the potential application of EMBR-seq+ on low-input or even single-cell samples, the authors should comment on the minimal input required for EMBR-seq+ to enable an efficient rRNA depletion while obtaining sufficient yield for sequencing.*

We thank the reviewer for this question. We are currently working on scaling down EMBR-seq+ to single cells. In our previous work, we showed that EMBR-seq can be used to efficiently deplete rRNA and sequence mRNA from 0.02 ng of total bacterial RNA (Wangsanuwat *et al.*, 2020). To address this comment, and comment 4 of Reviewer #1, we have added the following text to the Discussion section of the revised manuscript (lines 439-446):

“In addition, the use of blocking primers to deplete rRNA can potentially also be scaled down to single cells as we previously showed that EMBR-seq efficiently depletes rRNA from as little as 0.02 ng input total RNA (Wangsanuwat *et al.*, 2020). This corresponds to a pool of ~200 single bacteria (assuming ~0.1 pg of RNA per bacterial cell) and is in line with the sensitivity requirements of the existing methods that employ combinatorial barcoding strategies (Blattman *et al.*, 2020; Kuchina *et al.*, 2020; Wang *et al.*, 2022; Ma *et al.*, 2023). Therefore, we anticipate that in the future EMBR-seq+ could potentially be applied to single cells, thereby adding to the growing toolkit for profiling individual bacterial cells.”

4) *Figure 3F, Supplementary Figure S8: The DESeq2 analysis performed in the monoculture of F. succinogenes shows reproducible data across all replicates, whereas the replicates for the co-culture show clear differences in the gene expression profiles. The authors should comment on this.*

While the relative trends between monoculture and co-culture conditions are consistent, we agree with the reviewer that the biological replicates for the co-cultures show some variation. To ensure consistency, all the monoculture and co-culture replicates were prepared from the same seed cultures of *A. robustus*, *C. churrovii*, and *F. succinogenes* strain UWB7, and we followed the same experimental timeline for all the experiments. We hypothesize that the co-cultures exhibit more variation than the monocultures due to the increased complexity of the interactions between species that might depend on their relative abundances, and therefore small technical variations in the initial inoculation of bacteria and fungi may impact the final gene expression profiles observed.

5) *Supplementary Figure S9: The DESeq2 data for C. churrovii shows clear differences among replicates of monoculture as well as co-culture. Also at this point, the authors should comment on the discrepancies.*

The monocultures in Supplementary Figures S8 and S9 are the same samples. The variability between the monoculture samples is partly due to visual artifacts in the heatmap colors as a result of the Z-normalization across the 6 samples (across each row corresponding to one gene). Variability in the co-cultures could potentially arise due to reasons discussed in the comment above (response to Reviewer #3, comment 4). In the revised manuscript, we have now discussed the variability highlighted by the reviewer in this comment and the previous one (lines 339-345):

“While the relative trends between monoculture and co-culture conditions were consistent, we observed that the biological replicates for the co-cultures showed some variation, possibly due to the increased complexity of the interactions between species that might depend on their relative abundances. Therefore, to ensure consistency, all the monoculture and co-culture replicates were

prepared from the same seed cultures of *A. robustus*, *C. churrovis*, and *F. succinogenes* strain UWB7, and we followed the same experimental timeline for all the experiments.”

Minor points:

6) *For the investigated CAZyme, one or two descriptive sentences are required in the introduction. Also, the abbreviation CAZyme is not defined.*

We have now added the following details on CAZymes in the Introduction (lines 74-86):

“An important aspect of these microbial communities is understanding how different community members utilize different classes of CAZymes to efficiently deconstruct biomass. CAZymes are classified based on biochemical function or sequence homology into six broad families (glycoside hydrolase (GH), glycosyltransferase (GT), polysaccharide lyase (PL), carbohydrate esterase (CE), auxiliary activities (AA), and carbohydrate binding module (CBM)), and the comparative CAZyme expression profiles of community members can help inform the design of engineered communities for bio-based energy solutions (Lawson *et al.*, 2019; Drula *et al.*, 2022). Thus, more broadly, efficient sequencing of mRNA from non-model bacteria in microbial consortia will enable a deeper understanding of the role of each member within complex communities, and will provide opportunities to advance microbiome engineering for a variety of agricultural, environmental, or energy-related applications.”

The abbreviation CAZyme was previously defined in the introduction (line 69) and in the results section (line 238-239). In the revised manuscript, we have also included the abbreviation in the abstract (line 42-43).

7) *Lines 77-83: These statements require appropriate references.*

We have now included citations in this part of the manuscript (now lines 87-95).

8) *Lines 91-94: Supporting references are missing.*

We have now provided catalog numbers in Supplementary Table S1 so that researchers can easily find these commercial kits. The reference to Supplementary Table S1 is on line 102 in the revised manuscript.

9) *Line 269: to quantify.*

We thank the reviewer for pointing this out. We have corrected this grammatical error in the revised manuscript (line 301).

10) *Line 386: A major change that we introduced in EMBR-seq+ compared to EMBR-seq is the inclusion...*

We have added the phrase “compared to EMBR-seq” to the sentence in the revised manuscript (line 425).

11) *Lines 375-379, 434-436, and 439-440: These sections are highly redundant and can be shortened.*

To reduce redundancy, we have removed the last paragraph and updated the previous paragraph as follows (lines 482-485):

“Given the low cost, ease of design and efficiency of our method, we anticipate that high-throughput screens employing EMBR-seq+ will offer a strategy for rapidly understanding microbial interactions in diverse environmental systems.”

12) *Spelling of the species names in the references should be harmonized (only some are written in italics, e.g. lines 769, 808).*

We thank the reviewer for pointing out these corrections. We have included these changes in the revised manuscript (References section numbers: 5, 13, 26-28, 32, 38-41, 46, 58-59).

13) *Figure 3B and S4: The different conditions shown in this figure should also be indicated with the terms EMBR-seq and EMBR-seq+ to stay consistent. This also includes some clarification in the corresponding text (lines 239-248).*

We thank the reviewer for this comment. In the updated Figure 3B and Supplementary Figure S4, the pink box has been revised to “Unprocessed total RNA (EMBR-seq)”, the yellow box has been revised to “Poly-A depleted (EMBR-seq)”, and the green box has been revised to “Poly-A depleted and RNase H-treated (EMBR-seq+)”, such that it matches the labels in Figure 3A. We have accordingly also modified the Results, Methods, and figure captions as follows:

“Unprocessed total RNA (EMBR-seq)”: Main manuscript lines 260, 270, 1078, 1082, and Supplementary Information line 66.

“Poly-A depleted (EMBR-seq)”: Main manuscript lines 268, 273, 708, 712, 716, 1080, 1083, 1085, and Supplementary Information lines 67-68, and 68.

“Poly-A depleted & RNase H treated (EMBR-seq+)”: Main manuscript lines 284, 1081-1082, 1083, and Supplementary Information line 69.

14) *The replicate numbers shown for the DESeq2 analysis should be kept in order for all shown experiments (e.g. Figure S8, S9).*

The replicate order appears different in various panels because it is based on the distance between samples obtained from hierarchical clustering. We kept the ordering this way to more clearly indicate which samples are more closely related to each other for the gene set being considered in those panels. If the reviewer thinks it would be more helpful to show the heatmap with the same replicate order in all panels, we can remove the hierarchical clustering.

Supplementary Information:

15) *Line 62: libraries.*

We thank the reviewer for pointing this out. We have corrected this typographical error in the revised Supplementary Information (line 62).

September 12, 2023

Prof. Siddharth Subhas Dey
University of California Santa Barbara
Chemical Engineering and Biological Engineering
Santa Barbara, CA 93106

Re: mSystems00281-23R1 (Targeted rRNA depletion enables efficient mRNA sequencing in diverse bacterial species and complex co-cultures)

Dear Prof. Siddharth Subhas Dey:

I am pleased to inform you that your manuscript has been accepted, and I am forwarding it to the ASM Journals Department for publication. For your reference, ASM Journals' address is given below. Before it can be scheduled for publication, your manuscript will be checked by the mSystems production staff to make sure that all elements meet the technical requirements for publication. They will contact you if anything needs to be revised before copyediting and production can begin. Otherwise, you will be notified when your proofs are ready to be viewed.

If you would like to submit a potential Featured Image, please email a file and a short legend to msystems@asmusa.org. Please note that we can only consider images that (i) the authors created or own and (ii) have not been previously published. By submitting, you agree that the image can be used under the same terms as the published article. File requirements: square dimensions (4" x 4"), 300 dpi resolution, RGB colorspace, TIF file format.

We recognize that the video files can become quite large, and so to avoid quality loss ASM suggests sending the video file via <https://www.wetransfer.com/>. When you have a final version of the video and the still ready to share, please send it to mSystems staff at msystems@asmusa.org.

Sincerely,

Danielle Tullman-Ercek
Editor, mSystems

Journals Department
E-mail: mSystems@asmusa.org